# Bicc1 ribonucleoprotein complexes specifying organ laterality are licensed by ANKS6-induced structural remodeling of associated ANKS3

**Benjamin Rothé**[1], **Yayoi Ikawa**[2], **Zhidian Zhang**[3], **Takanobu A. Katoh**[2], **Eriko Kajikawa**[2], **Katsura Minegishi**[4], **Sai Xiaorei**[4], **Simon Fortier**[1], **Matteo Dal Peraro**[3], **Hiroshi Hamada**[2], **Daniel B. Constam**[1] *

**1** Ecole Polytechnique Fédérale de Lausanne (EPFL) SV ISREC, Lausanne, Switzerland, **2** Laboratory for Organismal Patterning, RIKEN Center for Biosystems Dynamics Research, Kobe, Japan, **3** Ecole Polytechnique Fédérale de Lausanne (EPFL) SV IBI, Lausanne, Switzerland, **4** Department of Molecular Therapy, National Institutes of Neuroscience, National Center of Neurology and Psychiatry (NCNP), Tokyo, Japan

* Daniel.Constam@epfl.ch

**Data Availability Statement:** All relevant data are within the paper and its Supporting Information files.

## Abstract

Organ laterality of vertebrates is specified by accelerated asymmetric decay of *Dand5* mRNA mediated by Bicaudal-C1 (Bicc1) on the left side, but whether binding of this or any other mRNA to Bicc1 can be regulated is unknown. Here, we found that a CRISPR-engineered truncation in ankyrin and sterile alpha motif (SAM)-containing 3 (ANKS3) leads to symmetric mRNA decay mediated by the Bicc1-interacting *Dand5* 3′ UTR. AlphaFold structure predictions of protein complexes and their biochemical validation by in vitro reconstitution reveal a novel interaction of the C-terminal coiled coil domain of ANKS3 with Bicc1 that inhibits binding of target mRNAs, depending on the conformation of ANKS3 and its regulation by ANKS6. The dual regulation of RNA binding by mutually opposing structured protein domains in this multivalent protein network emerges as a novel mechanism linking associated laterality defects and possibly other ciliopathies to perturbed dynamics in Bicc1 ribonucleoparticle (RNP) formation.

## Introduction

Cystic kidneys and left-right patterning defects are common features of ciliopathies caused by abnormal function of primary cilia or their downstream pathways [1]. During early development, stimulation of primary cilia by a leftward fluid flow accelerates the decay of *Dand5* mRNA specifically on the future left side of the node to thereby enable normal asymmetric patterning of surrounding primordial cells by the Nodal signaling pathway [2,3]. Recently, we and others showed that the asymmetric inhibition of *Dand5* mRNA downstream of flow-sensing cilia is mediated by the RNA-binding protein Bicaudal-C1 (Bicc1), both in mouse and xenopus [4,5]. In addition, loss of Bicc1 in vertebrates provokes the development of fluid-filled

**Funding:** This work was supported by the Human Frontiers Science Program fellowship LT000216/2016 to S.F., and by Rare Diseases GRS-051/13 grant from Gebert Rüf Stiftung to D.B.C. The funders had no role in study design, data collection and analysis, decision to publish, or preparation of the manuscript.

**Competing interests:** The authors have declared that no competing interests exist.

**Abbreviations:** 3AT, 3-aminotriazole; Adcy6, adenylate cyclase 6; AU, arbitrary unit; Bicc1, Bicaudal-C1; CPM, counts per million; EH, end-helix; GST, glutathione S-transferase; IVS, intervening sequence; KH, K-homology; KHL, KH-like; LPM, lateral plate mesoderm; ML, mid-loop; PBS, phosphate-buffered saline; PKD, polycystic kidney disease; RNP, ribonucleoparticle; RT-qPCR, reverse transcription quantitative polymerase chain reaction; SAM, sterile alpha motif; SD, standard deviation; WISH, whole mount in situ hybridization; Y2H, yeast-two-hybrid.

cysts in kidneys, pancreas, and hepatic bile ducts that are reminiscent of polycystic kidney diseases (PKDs) [6]. While no viable homozygous mutations have been identified in humans, heterozygous mutations associate with unilateral renal cystic dysplasia [7].

Bicc1 consists of a tandem repeat of 3 K-homology (KH) and 2 KH-like (KHL) domains that are separated from a self-polymerizing sterile alpha motif (SAM) domain by a disordered intervening sequence (IVS). The KH1 and KH2 domains of Bicc1 directly bind to a bipartite GAC motif in the conserved GACGUGAC sequence of the *Dand5* 3′ UTR that mediates the accelerated decay of *Dand5* mRNA upon flow stimulation of primary cilia on the left side of the node [5]. Functionally relevant stimuli of kidney cyst formation in PKD include cAMP [8]. Cyclic AMP also accumulates in Bicc1 mutant kidneys, correlating with de-repression of adenylate cyclase 6 (Adcy6) mRNA translation [9].

Silencing of *Adcy6* mRNA by Bicc1 depends on the KH domains to bind the 3′ UTR and on self-polymerization of the SAM domain to concentrate Bicc1 and target mRNAs in cytoplasmic granules [9,10]. Head-to-tail interaction of SAM domains is mediated by so-called mid-loop (ML) and end-helix (EH) surfaces that result in the ordered assembly of a large variety of homomeric or heteromeric complexes, ranging from dimers to extended polymers [11,12]. Known Bicc1-interacting factors include the ankyrin repeat and SAM-containing proteins ANKS3 and ANKS6 [13–17]. ANKS3 and ANKS6 mutations have been shown to perturb left-right patterning in a family with laterality defects or in animal models, respectively [18–20]. In addition, ANKS6 mutations associate with chronic kidney disease in nephronophthisis patients [13,18,19,21,22]. Bicc1 and ANKS3 bind each other via heterodimerization of their SAM domains and through contacts involving at least the KH domains [15,17]. Reconstitution experiments in heterologous cells revealed that the C-terminal domain of ANKS3 sterically hinders the elongation of Bicc1 polymers. By contrast, ANKS6 does not directly associate with either the SAM or with KH domains of Bicc1. However, the SAM domain of ANKS6 has a 10-fold higher affinity than Bicc1 to specifically bind the ANKS3 SAM domain [17,23]. By sequestering the ANKS3 SAM domain, ANKS6 liberates the Bicc1 SAM domain to self-polymerize in Bicc1-ANKS3-ANKS6 heterooligomers. However, whether and how remodeling of Bicc1 complexes by ANKS3 and ANKS6 impacts mRNA binding is unknown.

Here, we show that a targeted deletion in the murine *Anks3* gene leads to left-right patterning defects and ectopic down-regulation of the Bicc1-regulated *Dand5* 3′ UTR reporter. To directly test a role for ANKS3 as a Bicc1 antagonist, we reconstituted Bicc1 ribonucleoparticles (RNPs) with ANKS3 alone or together with ANKS6, and we mapped their multivalent protein interactions by combining AlphaFold structure predictions and yeast-two-hybrid (Y2H) and glutathione S-transferase (GST) pull-down assays. We show that Bicc1 binding to the *Dand5* 3′ UTR and other validated target mRNAs is antagonized by an interaction of the KH repeat with the C-terminal coiled coil domain of ANKS3. Conversely, co-recruitment of ANKS6 increases RNA binding by assisting ANKS3 to clamp down its coiled coil in an alternative conformation. Taken together, our data suggest that ANKS6 modulates the conformation of ANKS3-Bicc1 complexes to thereby license their recruitment to specific transcripts.

## Results

### A deletion in Anks3 leads to excessive mRNA decay by the Dand5 3′ UTR and randomizes organ laterality

Alternative splicing of ANKS3 can give rise to at least 5 protein-coding isoforms that share exons 9 and 10 (**S1A Fig**). To mutate all these ANKS3 isoforms, we deleted exons 10–11 using CRISPR/Cas9 editing (**S1B Fig**). Splicing of exon 9 directly to exon 12 thus can be expected to shift the reading frame and truncate ANKS3 after the Ank repeat.

Heterozygous $Anks3^{+/\Delta ex10\text{-}11}$ mice derived from targeted cells appeared phenotypically normal. However, intercrossing revealed that all homozygous mutants ($Anks3^{\Delta/\Delta}$) died before or immediately after birth, showing *situs* inversions (2/9) or heterotaxia (7/9) characterized by abnormal lung and liver lobation, cardiac malformations, and malpositioning of abdominal and thoracic vessels (**Fig 1A** and **S1 Table**). Whole mount in situ hybridization analysis at the 4-somite stage revealed that *Nodal* mRNA expression, which normally is up-regulated specifically on the left side, was bilaterally symmetric at the node ($n = 4/4$) and either lost ($n = 3/4$) or bilateral ($n = 1/4$) in the lateral plate mesoderm (LPM) of $Anks3^{\Delta/\Delta}$ embryos (**Fig 1B**). Moreover, while *Nodal* mRNA expression in left LPM of wild-type embryos ceases before the 7-somite stage, significant expression persisted in LPM on the left ($n = 1/2$) or on both sides ($n = 1/2$) of $Anks3^{\Delta/\Delta}$ embryos. To test whether *Anks3* controls left-right patterning by regulating Dand5, we intercrossed $Anks3^{+/\Delta}$ heterozygotes with the NDE-Hsp-dsVenus-*Dand5* 3′ UTR transgene. This reporter recapitulates *Dand5* mRNA decay and its regulation by flow-sensing cilia owing to the presence of a Bicc1-binding GACGUGAC motif in the 3′ UTR [5] (**Figs 1C** and **S1C**). Analysis in *Anks3* heterozygous and wild-type control embryos confirmed up-regulation of dsVenus fluorescence specifically in crown cells of the node, followed by down-regulation on the future left side during the 1- to 2-somite stage ($n = 12$). By contrast, dsVenus levels in $Anks3^{\Delta/\Delta}$ embryos were reduced 7.3-fold and remained low on both sides of the node ($n = 6$). Immunofluorescent staining revealed no detectable change in Bicc1 protein expression levels (**S1D Fig**). These results show that ANKS3 or at least its sterile α motif or the adjacent C-terminal coiled coil domain is essential to protect the *Dand5* 3′ UTR against bilaterally symmetric mRNA decay, possibly by inhibiting Bicc1.

## Bicc1 and ANKS3 are coexpressed with ANKS6 in node cells

To specify organ laterality by directly interacting with Bicc1 and/or ANKS6, ANKS3 would have to be expressed in the same cells at the node. In absence of available antibodies that could specifically detect endogenous ANKS3, we mined a public dataset of gene expression in single cells of early mouse embryos [24]. *Foxj1* and *Bicc1* are specifically expressed in node cells, starting at mid- or late-primitive streak stages, respectively [25–27]. We found that *Anks3*, *Anks6*, and *Dand5* are indeed transcribed in the *Foxj1*⁺/*Bicc1*⁺ cells (**S1E Fig**), as expected if ANKS3 and its ability to recruit ANKS6 regulate the function of Bicc1.

## ANKS3 binds an extended surface of more than one KH domains of Bicc1

Previous analysis by Y2H assay established that ANKS3 can bind the Bicc1 SAM domain and the KH repeat [17]. Therefore, to evaluate whether binding of one or several KH domains to mRNAs may be regulated, we first tried to map their contacts with ANKS3 using $KH_1$, $KH_2$, or a combination of $KHL_1$-$KH_3$-$KHL_2$ domains (Bicc1-KHL) as prey in Y2H assays. The strength of interaction was assessed by titrating 3-aminotriazole (3AT), a competitive inhibitor of the reporter gene product. We found that ANKS3 binding to the KH tandem repeat resisted up to 60 mM 3AT, whereas among KH fragments, only Bicc1-KHL bound ANKS3 above background levels, and this interaction was lost already at 0.5 mM 3AT (**Fig 2A**). For comparison, an intermediate concentration of 10 mM 3AT inhibits binding of ANKS3 to the Bicc1 SAM domain [17]. Thus, more than one KH and/or KHL domains of Bicc1 synergize to bind ANKS3 with high affinity through an interaction that is even stronger than the one between SAM:SAM heterooligomers. To further evaluate the contribution of individual KH domains by an independent approach, cell extracts containing ANKS3-Flag were incubated with Bicc1 GST fusion proteins on glutathione sepharose beads. Western blot analysis of bound proteins revealed that the complete KH tandem repeat retained ANKS3-Flag, whereas individual $KH_1$

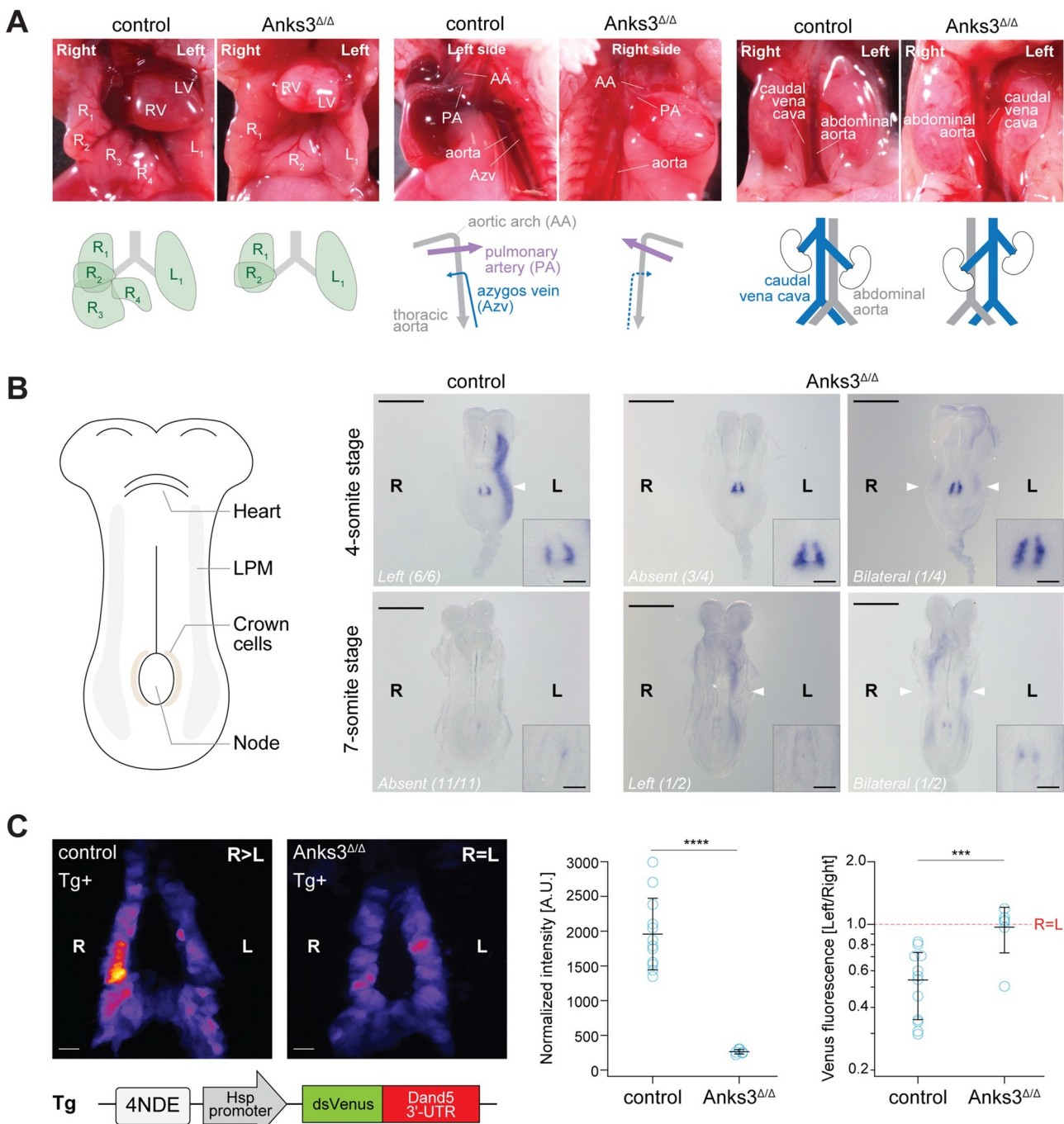

**Fig 1. Left/right patterning defects in Anks3^Δ/Δ mutant mice. (A)** Lungs, hearts, thoracic, and abdominal blood vessels of representative Anks3 ^Δ/Δ pups and heterozygous control litter mates on postnatal day P0. Schematic depictions of organ patterning are shown below. R, L: right and left pulmonary lobes, respectively. **(B)** Schematic ventral view of a mouse embryo at E8.0 (left) and WISH analysis of *Nodal* mRNA expression in *Anks3*^Δ/Δ embryos and wild-type control litter mates at 4- and 7-somite stages (right). Scale bars, 500 μm or 100 μm for the magnified node regions (insets). The Nodal pattern (left, bilateral, or absent) in LPM and the number of embryos corresponding to each phenotype are indicated. **(C)** dsVenus fluorescence at the node of *Anks3*^Δ/Δ and wild-type control embryos harboring the NDE-Hsp-dsVenus-*Dand5* 3′ UTR transgene. The dsVenus coding sequence fused to the DNA sequence for the 3′ UTR of mouse Dand5 mRNA is under the control of the mouse Hsp68 promoter and 4 copies of the crown cell-specific enhancer (NDE) of mouse *Nodal*. Images are representative of at least 6 embryos per genotype at the 3- to 5-somite stage. The graphs represent the global dsVenus intensity and fluorescence ratios on the right versus left side of the node for each genotype. Blue circles indicate individual values. Data are means ± SD. Statistical significance was determined using the Mann–Whitney U test, with ns: nonsignificant, ***$p < 0.001$, ****$p < 0.0001$. Scale bar, 50 μm. Underlying data can be found in the S1 Raw Values file. LPM, lateral plate mesoderm. WISH, whole mount in situ hybridization.

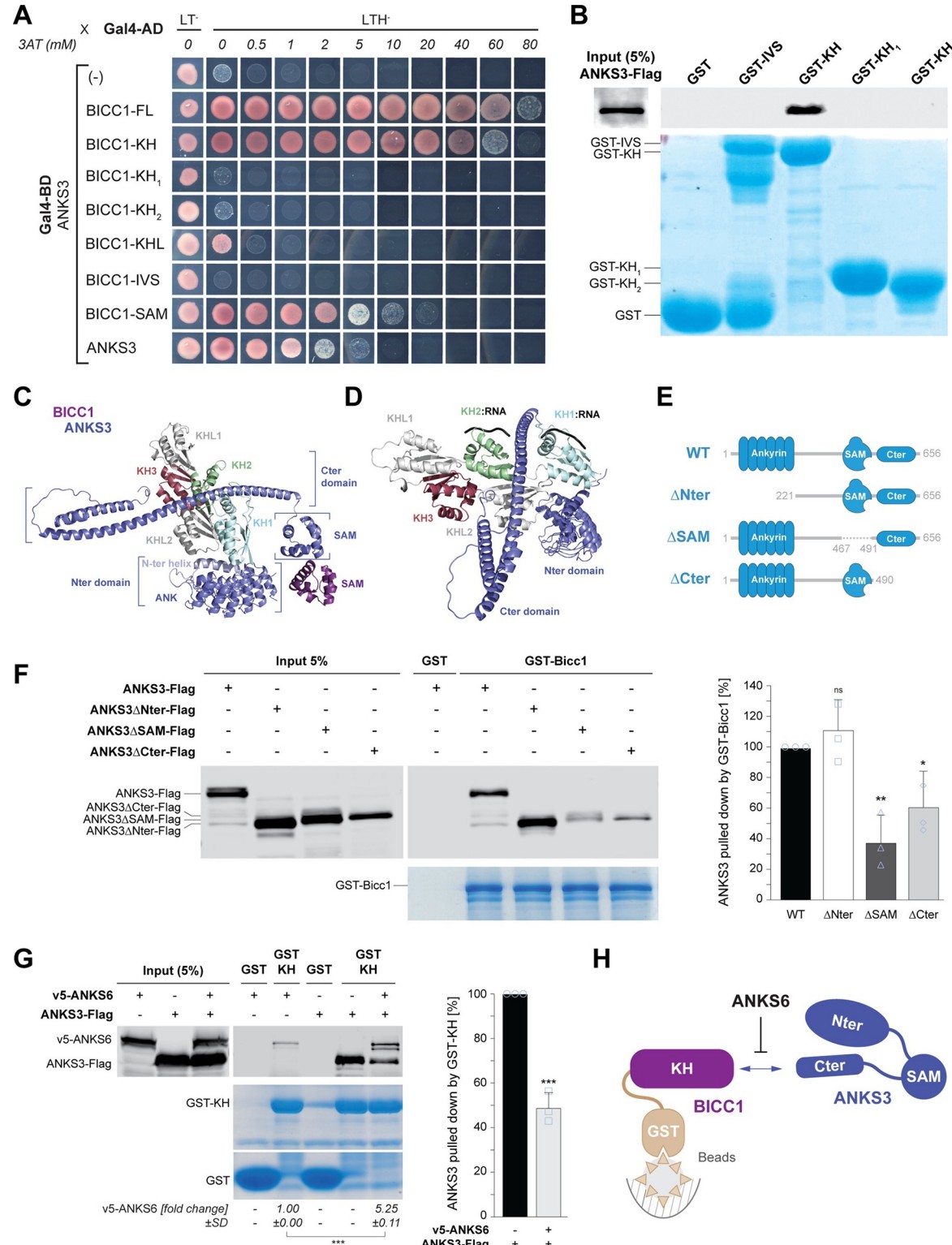

**Fig 2. ANKS3 binds to the Bicc1 KH domains and competes with RNA binding. (A)** Yeast two-hybrid mapping of the interaction between a fusion of ANKS3 with the DNA-binding domain of Gal4 (Gal4-BD) and human BICC1 KH domains fused to Gal4 activation domain (Gal4-AD). Controls in nonselective medium without leucine (L) and tryptophan (T) are shown in the first column (LT⁻). Interactions were revealed at the indicated concentrations of 3AT in triple selective medium (LTH⁻) lacking histidine. **(B)** Pull-down of ANKS3-Flag from HEK293T cell extracts by glutathione sepharose beads coated with recombinant domains of Bicc1 fused to GST.

**(C, D)** 3D model of structured Bicc1 interfaces with ANKS3 predicted by AlphaFold viewed from above the RNA-binding KH domain surfaces (C) or sideways (D). The structured domains are annotated. For the sake of clarity, intrinsically disordered regions are not shown. Note the interaction between the Bicc1 KH domains and a C-terminal coiled coil of ANKS3 (Cter). **(E)** Cartoon depicting the domain organization of full-length ANKS3 and its truncated variants. **(F)** Left: Pull-down of full-length and truncated ANKS3-Flag in HEK293T cell extracts by GST-Bicc1 or GST alone (control). Right: Quantification of bound ANKS3-Flag (100%) and of its indicated truncation mutants. **(G)** Left: pull-down of ANKS3-Flag in HEK293T cell extracts by GST-Bicc1-KH in presence or absence of v5-ANKS6. The amounts of v5-ANKS6 that were pulled down from the HEK293T cell extracts are quantified below. Right: Quantification of bound ANKS3-Flag normalized to the pull-down without ANKS6 (100%). **(H)** Cartoon depicting the protein–protein interactions observed in panel G. Data are means + SD from 3 independent experiments. ns: nonsignificant, $^*p < 0.05$, $^{**}p < 0.01$, $^{***}p < 0.001$ (Student's $t$ test). Underlying data can be found in the S1 Raw Images and S1 Raw Values files. 3AT, 3-aminotriazole; Bicc1, Bicaudal-C1; GST, glutathione S-transferase; KH, K-homology; SAM, sterile alpha motif.

or $KH_2$ domains or the IVS did not (**Fig 2B**). GST fusions of $KH_3$ with or without the KH-like 1 or 2 domains could not be tested due to their insolubility. Nevertheless, these results corroborate our Y2H data that tight binding of Bicc1 to ANKS3 requires an extended surface of more than one KH and/or KHL domains.

## Bicc1 associates with a C-terminal coiled coil of ANKS3

To identify which regions of ANKS3 bind Bicc1 KH domains and how their conformation might be affected by SAM:SAM interactions and by ANKS6, we modeled the structure of these protein complexes using the AlphaFold Multimer algorithm [28]. AlphaFold predicts that ANKS3 consists of an α-helix and a repeat of 6 Ank domains at the N-terminus (hereafter named Nter), followed by a disordered region, the SAM domain and a C-terminal coiled coil domain (Cter) (**S2 Fig**). Interestingly, in complexes of Bicc1 with ANKS3 alone, the ANKS3 Cter is inserted on top of $KHL_2$ between the RNA-binding surfaces of the $KH_1$ and $KH_2$ domains of Bicc1 (**Fig 2C and 2D**). This conformation appears to be favored by SAM domain heterodimerization that anchors the coiled coil of ANKS3 in a defined orientation and thereby limits its freedom. To test these predictions, we first assessed which domains of ANKS3 mediate binding to full-length GST-Bicc1 fusion protein in pull-down assays. Deletion of the SAM or Cter domains of ANKS3-Flag diminished binding to GST-Bicc1 by 60% and 40%, respectively, whereas deletion of the Nter tended to increase it (**Fig 2E and 2F**). Together with our Y2H data and the model structure of Bicc1-ANKS3 complexes, these results indicate that multivalent contacts between these proteins function additively, and that they include a surface of several Bicc1 KH domains and the coiled coil of ANKS3.

## The N-terminal region of ANKS3 cooperates with ANKS6 to dislodge the ANKS3 coiled coil from Bicc1 KH domains

To validate the prediction of the structural model that the coiled coil of ANKS3 directly binds the Bicc1 KH domains (**Fig 2C**), we first tested whether ANKS3 can be pulled down by GST fused to KH domains alone, thereby eliminating the confounding contribution from the Bicc1 SAM domain. As shown in **Fig 2G**, GST-KH readily pulled down ANKS3-Flag from HEK293T cell extracts. Interestingly, co-transfection of v5-ANKS6 decreased this interaction by 50% and without changing ANKS3-Flag protein levels in the crude extracts that served as input. In turn, ANKS3-Flag increased the pull-down of v5-ANKS6 more than 5-fold above baseline. Besides confirming that ANKS3 mediates ANKS6 co-recruitment [17], these data show that ANKS6 in turn reduces a novel interaction of ANKS3 with the Bicc1 KH domains (**Fig 2H**).

To investigate the mechanism how ANKS6 liberates Bicc1 KH domains from ANKS3, we analyzed AlphaFold structure predictions of Bicc1-ANKS3-ANKS6 complexes. ANKS6 contains no coiled coil after its SAM domain, but an N-terminal repeat of 11 Ank domains (**S2**

Fig). As shown in Fig 3A, ANKS6, together with the Ank repeat of ANKS3, is predicted to clamp down the ANKS3 Cter beneath the KH domains of Bicc1. This model structure strongly suggests that ANKS6 dislodges ANKS3 from Bicc1 KH by clamping down its coiled coil. To verify that this clamp exists and that it involves Ank domains of ANKS3, we repeated the GST-KH pull-down using N-terminally truncated ANKS3-Flag (ANKS3 ΔNter). Compared to full-length ANKS3, binding of ANKS3 ΔNter to GST-KH increased 8-fold, regardless of the presence or absence of v5-ANKS6 (Fig 3B). Co-recruitment of v5-ANKS6 simultaneously decreased by 61%. Most importantly, the residual 39% of bound v5-ANKS6 failed to weaken the interaction of ANKS3 ΔNter with GST-KH. These results show that ANKS6 cooperates with the ANKS3 Nter to dislodge the ANKS3 Cter from Bicc1 KH domains.

## Hetero-oligomerization with Bicc1 via SAM:SAM interactions helps ANKS3 to compete with RNA binding in cell-free assays

AlphaFold predicts that ANKS3 might obstruct the RNA binding surface of the KH domains. However, conditions to unmask such an effect in vivo have remained elusive [17]. To assess

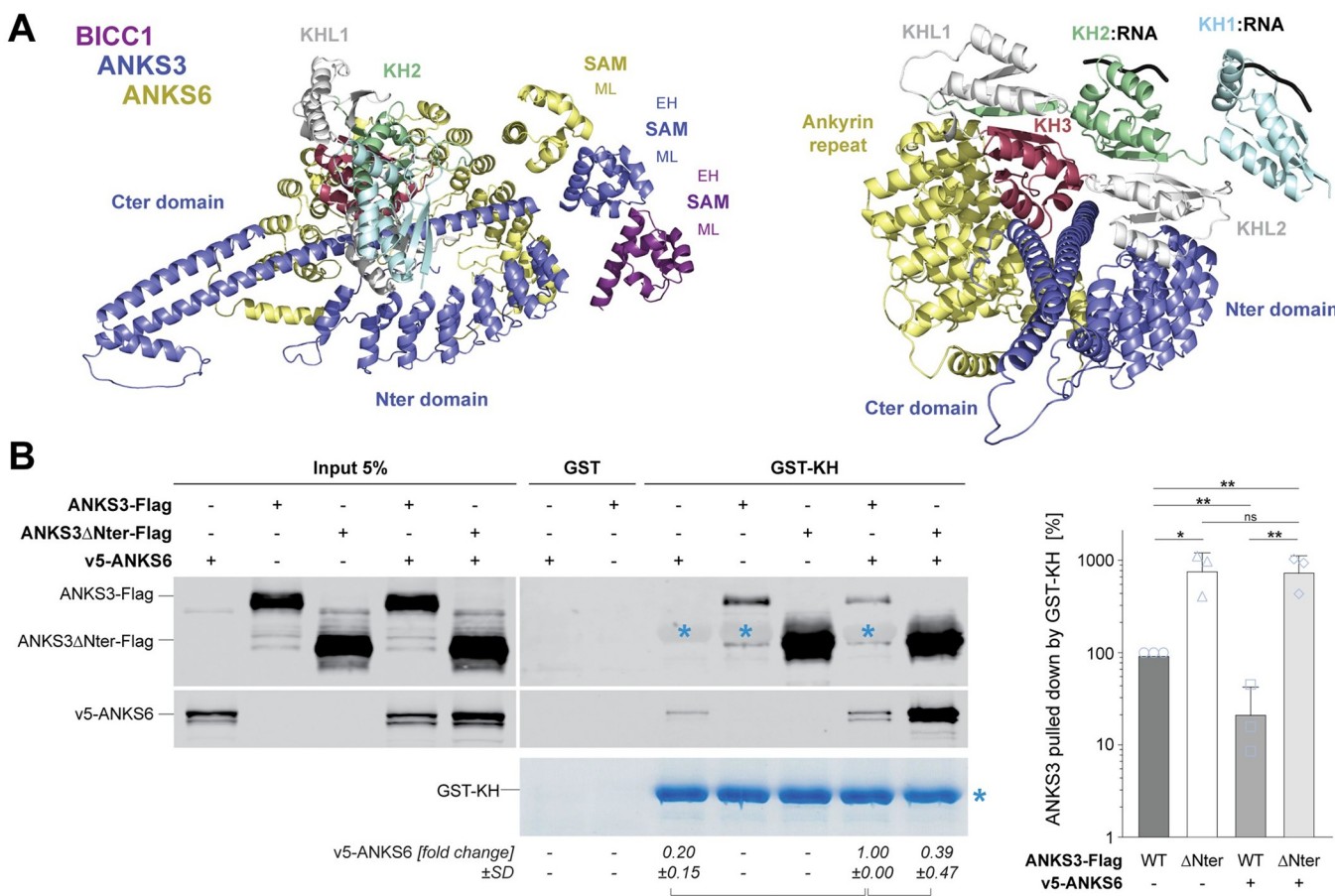

**Fig 3. ANKS6 destabilizes the ANKS3-KH interaction by displacing the ANKS3 coiled coil. (A)** Structure of Bicc1-ANKS3-ANKS6 complexes predicted by AlphaFold, viewed from above the RNA-binding KH domain surfaces (left) or from the side (right). The structured domains are annotated. For the sake of clarity, intrinsically disordered regions are not shown. **(B)** Left: Pull-down of ANKS3-Flag full-length or ΔNter alone, or together with v5-ANKS6 by GST-KH. Note that GST-KH was so abundant in bead eluates that it is faintly visible also in the anti-Flag western blot as a shady band closely above ANKS3 ΔNter marked by a blue asterisk. The amounts of v5-ANKS6 relative to ANKS3-Flag that were pulled down from the HEK293T cell extracts are quantified below. Right: Quantification of GST-KH binding to the indicated ANKS3 truncation mutants versus full-length ANKS3-Flag (100%). Data are means + SD from 3 independent experiments. ns: nonsignificant, $^*p < 0.05$, $^{**}p < 0.01$, $^{***}p < 0.001$ (Student's $t$ test). Underlying data can be found in the S1 Raw Images and S1 Raw Values files. Bicc1, Bicaudal-C1; EH, end-helix; GST, glutathione S-transferase; KH, K-homology; ML, mid-loop; SAM, sterile alpha motif.

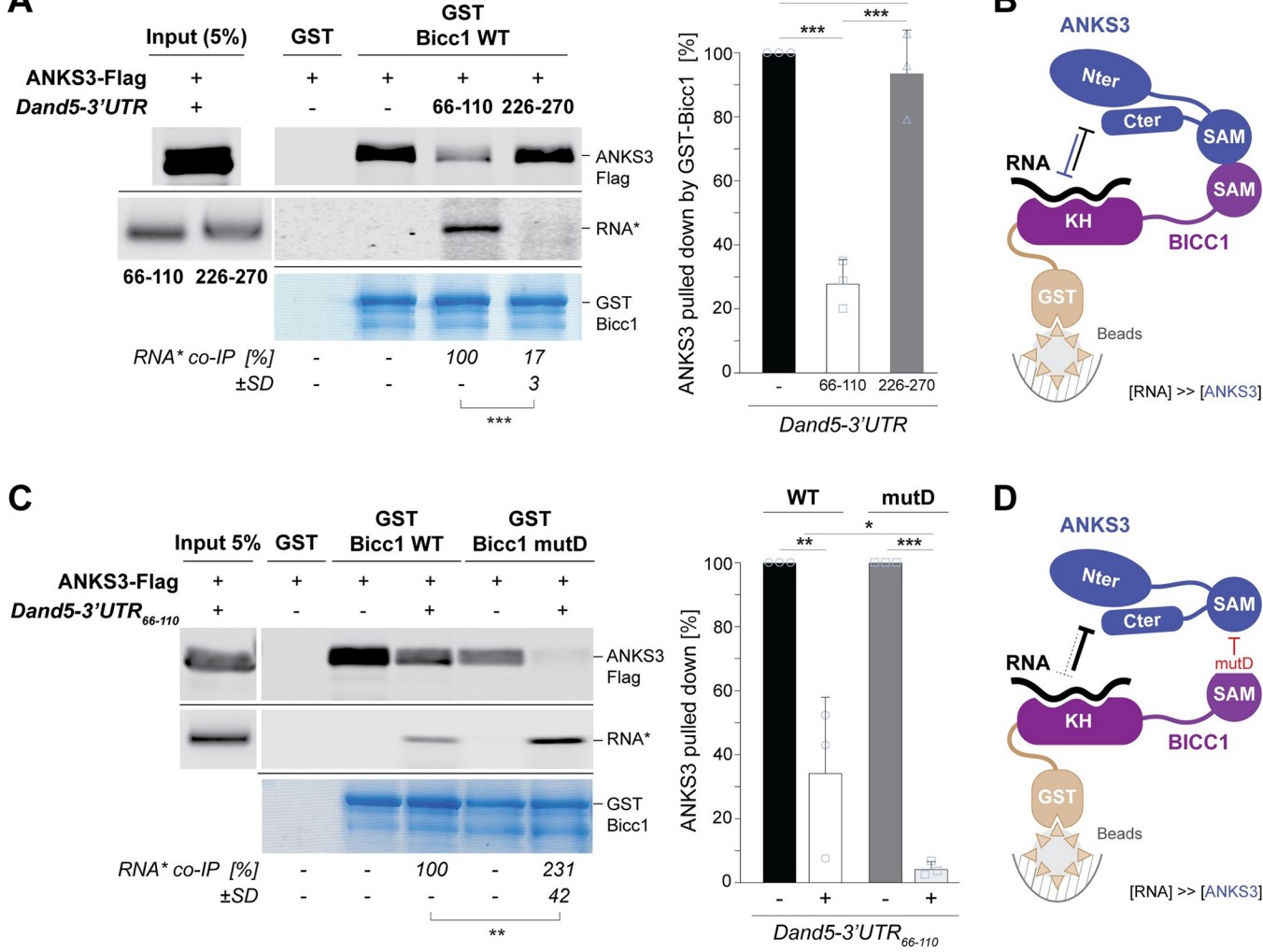

**Fig 4. ANKS3 and a specific target RNA compete for Bicc1 binding in vitro.** (A) Western blot analysis of ANKS3-Flag from HEK293T cell extracts before (input) and after pull-down by in vitro reconstituted RNPs of recombinant GST-Bicc1 that were pre-assembled with saturating amounts of the fluorescently labeled Dand5-3′ UTR RNA fragment 66–110 or, as a control for nonspecific binding, fragment 226–270 (*). Coomassie blue staining of GST-Bicc1 protein retained by the beads (bottom panel) and the fluorescence of bound RNA (middle panels) are shown below. The quantification of the amounts of target RNA (*) retained by the GST-Bicc1 beads before and after incubation with ANKS3-Flag is shown below the gels. The values for the amounts of pulled down ANKS3-Flag shown in the histogram to the right were normalized to controls without target RNA (100%). (B) Cartoon summarizing the result shown in panel A. (C) Western blot analysis of ANKS3-Flag before (input) and after pull-down by the SAM polymerization mutant GST-Bicc1 mutD versus WT GST-Bicc1 that were saturated with fluorescently labeled Dand5-3′ UTR RNA fragment 66–110 as in (A). Pull-downs of RNA and ANKS3-Flag were quantified as in (A). Values for the amount of bound RNA are relative to the pull-down by WT GST-Bicc1 (100%). (D) Cartoon summarizing the result shown in panel C. Data are means + SD from 3 independent experiments. ns: nonsignificant, *$p < 0.05$, **$p < 0.01$, ***$p < 0.001$ (Student's $t$ test). Underlying data can be found in the S1 Raw Images and S1 Raw Values files. GST, glutathione S-transferase; KH, K-homology; RNP, ribonucleoparticle; SAM, sterile alpha motif; WT, wild-type.

whether ANKS3 and RNA compete for Bicc1 binding in vitro, we performed GST pull-down in the presence of nucleotides 66–110 of the Dand5-3′ UTR RNA that bind KH1 and KH2 domains with high affinity (Kd = 200 nM) or with nucleotides 226–270 as a control RNA of similar size [5]. We found that preincubation of GST-Bicc1 with the 66–110 transcript inhibited the pull-down of ANKS3-Flag by 70% (**Fig 4A and 4B**). Analogous treatment with the 226–270 control RNA had no significant effect, indicating specificity. Since ANKS3 and Bicc1 also interact via their SAM domains [17], we wondered whether the SAM:SAM interface influences the competition. To test this, we repeated the ANKS3-Flag pull-down assay using a GST

fusion of Bicc1 mutD that lacks the SAM:SAM interface [10]. Interestingly, preincubation of GST-Bicc1 mutD with Dand5-3′ UTR$_{66-110}$ decreased the ANKS3-Flag pull-down to nearly background levels (**Fig 4C and 4D**). Conversely, the retention of RNA by mutD versus WT Bicc1 increased more than 2-fold. These data show that binding of Bicc1 to ANKS3 or RNA is mutually inhibitory and that SAM:SAM interactions bias this competition in favor of ANKS3.

## Endogenous ANKS3 buffers the binding of Bicc1 to specific target mRNAs in IMCD3 cells

To test whether endogenous ANKS3 and RNA compete for Bicc1 binding in cells, we analyzed Bicc1 RNPs by RNA co-immunoprecipitation in IMCD3 cells expressing doxycycline-inducible *Anks3* shRNA [29]. Reverse transcription (RT) and quantitative polymerase chain reaction (qPCR) analysis revealed only exceedingly low amounts of *Dand5* mRNA in these cells (**S3A Fig**). Therefore, we analyzed if *Anks3* knockdown alters the binding of endogenous Bicc1 to its own transcripts or to endogenous *Adcy6* mRNA, two alternative known targets in mammalian cells [9], and/or their expression. RT-qPCR analysis validated that doxycycline administration depleted *Anks3* mRNA by >80% (**Fig 5A**). While the *Adcy6* mRNA expression level remained unchanged, *Anks3* knockdown led to an unexpected 7.4-fold increase in *Bicc1* mRNA expression (**S3B Fig**), but no corresponding increase in Bicc1 protein (**Fig 5B**). It seemed possible, therefore, that *Bicc1* transcripts under these conditions are stabilized in a silenced form in

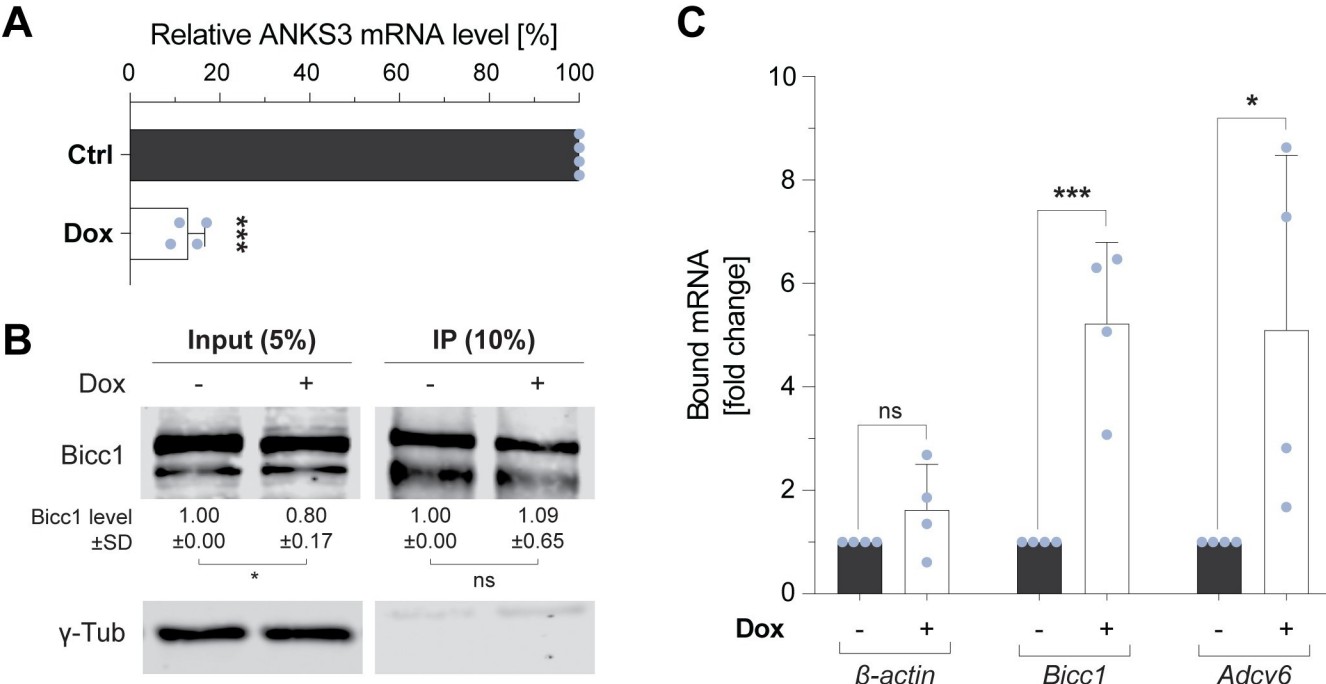

**Fig 5. ANKS3 stimulates the RNA-binding activity of endogenous Bicc1 in IMCD3 cells. (A)** RT-qPCR analysis of endogenous *Anks3* mRNA in IMCD3 cells treated with or without 0.25 μg/ml doxycycline (Dox) to induce the expression of *Anks3* shRNA. The levels of *Anks3* relative *to β-actin* mRNAs is expressed as a percentage of the baseline in untreated cells. **(B)** Western blot of endogenous Bicc1 in cytoplasmic extracts (inputs) and immunoprecipitates of IMCD3 cells treated with or without doxycycline to induce *Anks3* shRNA. γ-tubulin was a loading control. The levels of Bicc1 protein are expressed as a fold change of the baseline in untreated cells. For the inputs, the fold change was calculated after normalization to the γ-tubulin signal that served as internal control. **(C)** RT-qPCR analysis of the indicated mRNAs in IMCD3 before and after *Anks3* depletion in IMCD3 cells. The values are expressed as the fold change normalized to the untreated condition. The y-axis represents the amount of mRNA in Bicc1 immunoprecipitates normalized to Bicc1 protein in the IP fraction. Data are means + SD from 4 independent experiments. ns: nonsignificant, *$p < 0.05$, **$p < 0.01$, ***$p < 0.001$ (Student's *t* test). Underlying data can be found in the S1 Raw Images and S1 Raw Values files. Bicc1, Bicaudal-C1; RT-qPCR, reverse transcription quantitative polymerase chain reaction.

complexes with Bicc1 protein. To examine whether the RNA-binding capacity of Bicc1 is enhanced upon *Anks3* depletion, we compared the overall amounts of co-immunoprecipitated transcripts between *Anks3*-depleted cells and control cells. We found that *Anks3* knockdown increased the co-immunoprecipitation of both *Bicc1* and *Adcy6* transcripts on average 5-fold, whereas the binding to β-actin control mRNA was largely unchanged (**Fig 5B**). These results show that endogenous ANKS3 attenuates Bicc1 binding to both *Adcy6* and *Bicc1* transcripts.

## Gain-of-function experiments confirm that ANKS3 inhibits binding of Bicc1 to specific target mRNAs

To test whether ANKS3 similarly inhibits Bicc1 binding to mRNA in gain-of-function experiments, we co-transfected HA-Bicc1 and dsVenus-*Dand5* 3′ UTR reporter with ANKS3-Flag or empty vector control in HEK293T cells [5]. RT-qPCR analysis confirmed that unlike *β-actin* control mRNA, the dsVenus-*Dand5* 3′ UTR mRNA is enriched more than 28-fold on average in HA-Bicc1 immunoprecipitates relative to the background in cells without HA-Bicc1 (**S4A Fig**). Importantly, co-expression of ANKS3-Flag reduced the co-immunoprecipitation of this reporter mRNA with HA-Bicc1 by 75% (**Figs 6A and S4A**). ANKS3-Flag similarly inhibited binding of HA-Bicc1 to its own mRNA (**Fig 6A and 6B**) and without altering HA-*Bicc1* mRNA expression levels (**S4B Fig**). These results confirm that ANKS3 attenuates mRNA binding of Bicc1 and without perturbing the expression of HA-Bicc1 itself.

## The N-terminal region of ANKS3 synergizes with ANKS6 to license mRNA binding of the Bicc1 KH domains

Structure modeling and GST pull-downs assays pointed to a possible antagonism between the effects of Nter and Cter regions of ANKS3 on the RNA binding activity of Bicc1. To evaluate this model, we analyzed the effect of ANKS3 truncation mutants on Bicc1 RNP formation with its own transcripts or with co-expressed *Dand5* 3′ UTR reporter in RNA co-immunoprecipitation experiments. Interestingly, Bicc1 binding to these target mRNAs sharply increased in cells expressing ANKS3 ΔCter versus full-length ANKS3, along with a drastic reduction of Bicc1-ANKS3 association (**Fig 6A**). By contrast, coexpression with ANKS3 ΔNter largely abolished RNA binding, reducing it 3- to 5-fold below the baseline level seen with WT ANKS3 (**Fig 6A and 6B**). Since ANKS3 ΔCter protein was less expressed, and to rule out the possibility that it poorly inhibited Bicc1 RNP formation due to inefficient expression, we increased the dosage of transfected ANKS3 ΔCter-Flag by 2- or 4-fold. Importantly, transfection at a 4-fold elevated dosage enriched ANKS3 ΔCter in the co-immunoprecipitation to a similar extent as ANKS3 ΔNter. Despite this increase, ANKS3 ΔCter still failed to impair HA-Bicc1 binding to the *Dand5*-3′ UTR and instead even stimulated the recruitment of the *HA-Bicc1* transcript above the baseline without ANKS3. Together, these results suggest that whereas access of KH domains to mRNA is obstructed by the ANKS3 Cter, the Nter domain likely regulates this interaction.

To investigate how KH domains are liberated from ANKS3, we tested whether binding of Bicc1-ANKS3 complexes to specific transcripts can be induced by ANKS6. Co-expression of v5-ANKS6 depleted ANKS3-Flag in Bicc1 immunoprecipitates by 30% (**Fig 6B**), in line with our observation that ANKS6 co-recruitment similarly destabilized the association of ANKS3 with Bicc1 KH domains in cell-free reconstitution assays (**Fig 2G**). Concomitantly, v5-ANKS6 restored significant binding of HA-Bicc1 to the *Dand5* reporter mRNA and to HA-Bicc1 transcripts. In sharp contrast, in absence of the ANKS3 Nter, v5-ANKS6 was incapable of licensing the association of HA-Bicc1 with its own transcript and with the *Dand5* reporter mRNA. Together, these results suggest that binding to the ANKS3 Cter specifically obstructs the access

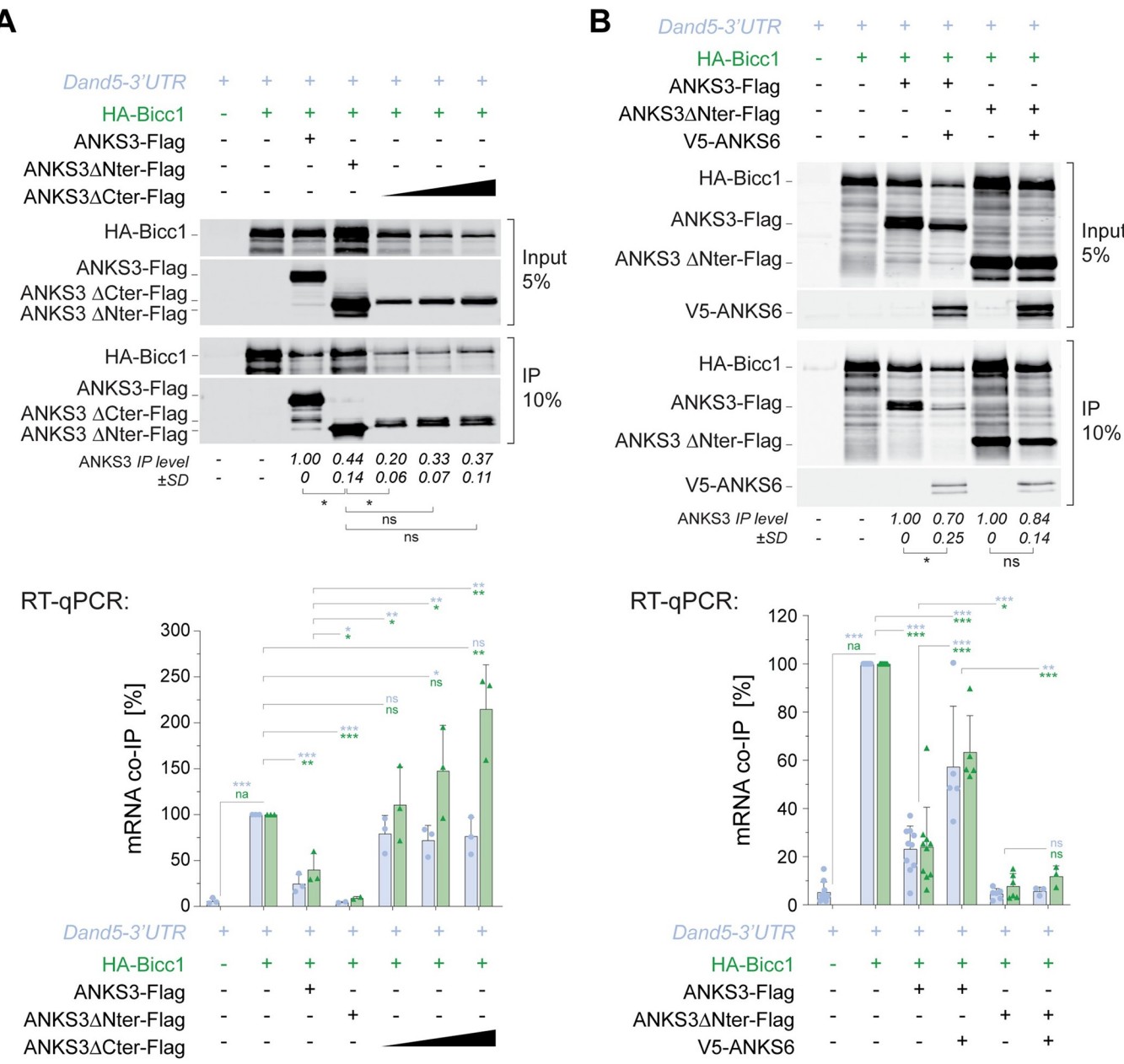

**Fig 6. ANKS6 augments Bicc1 binding to target mRNAs by cooperating with the N-terminal domain of ANKS3. (A)** Top: Representative western blots of the protein fractions in RNA co-immunoprecipitates from cytoplasmic extracts of HEK293T cells expressing the dsVenus-*Dand5*-3′ UTR reporter and HA-Bicc1 alone or in combination with full-length or truncated ANKS3-Flag. Three transfection doses (×1, ×2, and ×4) were used for ANKS3 ΔCter. The amounts of soluble ANKS3-Flag relative to HA-Bicc1 in the IP fractions are quantified below the blots. Bottom: Below the immunoblots, RT-qPCR analysis shows the ratios of co-immunoprecipitated *Dand5*-3′ UTR reporter mRNA and *HA-Bicc1* transcript normalized to their amounts in inputs and to HA-Bicc1 protein in the IP, relative to the control HA-Bicc1 IP without ANKS3 (100%). **(B)** Top: Representative western blots of the protein fractions in RNA co-immunoprecipitates from cytoplasmic extracts of HEK293T cells expressing the dsVenus-*Dand5* 3′ UTR reporter and HA-Bicc1 alone or in combination with ANKS3-Flag, or with its truncated form (Nter), and with or without v5-ANKS6. The amounts of soluble ANKS3-Flag relative to HA-Bicc1 in the IP fractions are quantified below the blots. Bottom: The RT-qPCR analysis shown below indicates the ratios of co-immunoprecipitated *Dand5*-3′ UTR reporter mRNA and *HA-Bicc1* transcript normalized to their amounts in inputs and to HA-Bicc1 protein in the IP, relative to the control HA-Bicc1 IP without ANKS3 (100%). Data are means + SD from at least 3 independent experiments. ns: nonsignificant, *$p < 0.05$, **$p < 0.01$, ***$p < 0.001$ (Student's *t* test). Underlying data can be found in the S1 Raw Images and S1 Raw Values files. Bicc1, Bicaudal-C1; RT-qPCR, reverse transcription quantitative polymerase chain reaction.

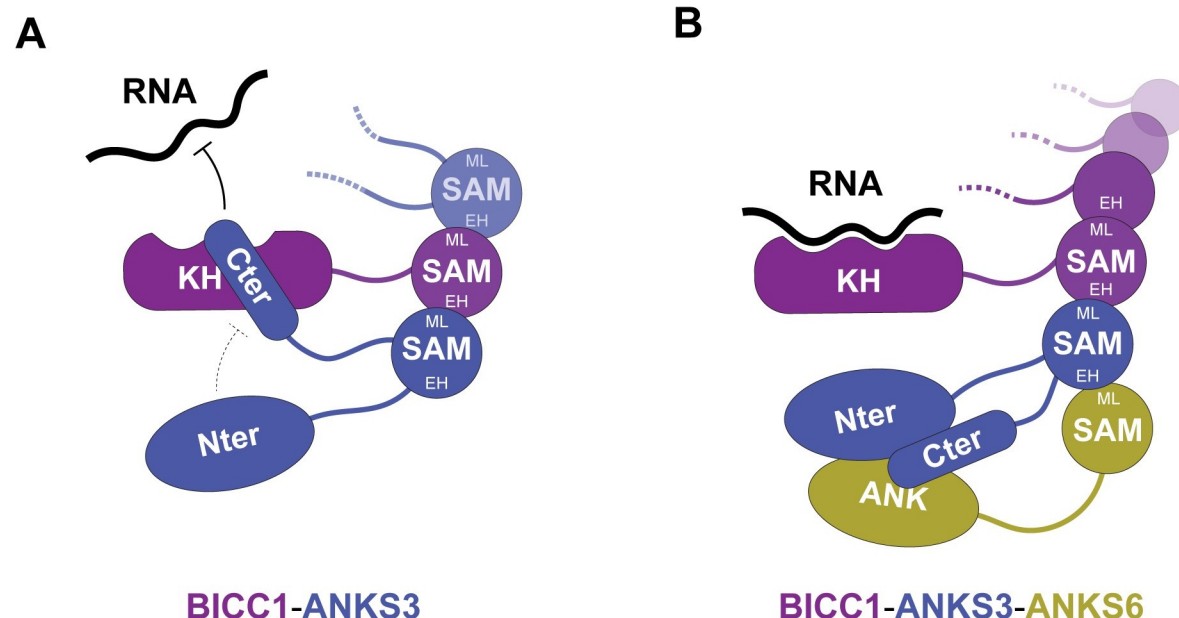

**Fig 7. Model of the licensing of Bicc1 RNP formation by ANKS3 and ANKS6. (A)** In absence of ANKS6, ANKS3 interacts with both ML and EH surfaces of the Bicc1 SAM domain. In parallel, the coiled coil of the ANKS3 Cter associates with the KH domains and inhibits RNA binding, while the Nter tends to attenuate this effect. **(B)** The incorporation of ANKS6 induces a topological remodeling of the complex. Due to a 10-fold higher affinity, ANKS6 hijacks the EH surface of the ANKS3 SAM domain [17,23]. In parallel, ANKS6 cooperates with the ANKS3 Nter to clamp down the coiled coil and to license RNA binding. Bicc1, Bicaudal-C1; EH, end-helix; KH, K-homology; ML, mid-loop; SAM, sterile alpha motif.

of KH domains to mRNA (**Fig 7A**). Conversely, cooperation of the ANKS3 Nter with ANKS6 overcomes this inhibition by clamping down the ANKS3 Cter in an alternative position (**Fig 7B**).

## Discussion

ANKS3 recruits Bicc1 and ANKS6 into a joint multivalent protein network in vitro and in mouse kidneys [13–17], and loss of either of these ANKS3-interacting factors results in organ laterality defects [18–20,27]. However, whether ANKS3 influences the binding of Bicc1 to the *Dand5* 3′ UTR or any other target mRNAs and conditions to reconstitute and analyze such a regulatory switch in vitro remained elusive. Here, a CRISPR-engineered mutation in mouse embryos that truncates ANKS3 after its Ank repeat led to impaired organ laterality and bilaterally symmetric degradation of the *Dand5* 3′ UTR reporter of Bicc1 activity. Structure–function analysis revealed that ANKS3 competes with target mRNAs to bind the KH domains. Conversely, co-recruitment of ANKS6 rescued mRNA binding through conformational remodeling of Bicc1-ANKS3 complexes. Mutual competition by structured domains is a new paradigm to license binding of a multivalent protein network to mRNA that suggests associated ciliopathies are likely linked to perturbed RNP assembly.

The *Dand5* 3′ UTR and its binding to Bicc1 are conserved in vertebrates and essential to specify the future left side in response to flow, but how flow stimulation of primary cilia activates Bicc1 is unknown [4,5]. To address this, we here prioritized to test whether Bicc1 RNA binding is regulated by known interacting factors such as ANKS3 and ANKS6 that are implicated in left-right patterning in humans or rodent models. To probe the role of ANKS3 in vivo, we deleted exons 10 and 11 that are shared by all splice variants described in mice. This

deletion truncates ANKS3 before the SAM domain. None of the 5 commercially available antibodies that we could test specifically detected ANKS3. Thus, the localization of wild-type ANKS3 and potential residual accumulation of truncated protein comprising the Ank repeat in $Anks3^{\Delta/\Delta}$ mice remain to be investigated. This caveat aside, mining of public scRNA-seq data confirmed that $Anks3$ and $Anks6$ transcripts are detected in mesendoderm, including in cells that coexpress $Foxj1$ and $Bicc1$ mRNAs, 2 independent markers of ciliated node cells [25–27]. Moreover, the randomization of left-right patterning observed in $Anks3^{\Delta/\Delta}$ mouse embryos and its recessive nature indicate a loss-of-function, consistent with the laterality defects in a family with mutations in human $ANKS3$, and with the randomization of heart looping described in $anks3$ zebrafish morphants [20].

Mechanistically, our analysis of the dsVenus-$Dand5$ 3′ UTR reporter transgene in $Anks3^{\Delta/\Delta}$ embryos revealed an essential function of ANKS3 to antagonize the mRNA decay that is induced by the Bicc1-interacting $Dand5$ 3′ UTR in crown cells of the node. Since Bicc1 expression at the node did not change, and since the decay of its target increased in $Anks3^{\Delta/\Delta}$ mutants, it follows that Bicc1 remained active. While these findings do not rule out additional unknown functions, they support a role of ANKS3 in inhibiting Bicc1 activity. The similarity of left-right patterning defects of $Anks3^{\Delta/\Delta}$ mutants (this study) and in $Bicc1^{-/-}$ mutants [5,27] is consistent with this model, because both excessive decay or ectopic accumulation of the $Dand5$ transcript randomize the sidedness of asymmetric Nodal signaling [30,31]. To further test this model in vivo, we considered to image the dsVenus-$Dand5$ 3′ UTR reporter and its response to artificial flow also in $Anks3; Bicc1$ double mutants. However, the huge number of mutant embryos and control litter mates required would not be justified. To conclusively assess whether ANKS3 protects this and possibly other target mRNAs from Bicc1-mediated decay, we instead directly tested its effect on Bicc1 RNP formation. Our in vitro reconstitution of Bicc1 RNPs with ANKS3 alone or together with ANKS6 combined with protein structure modeling revealed that ANKS3 directly competes with mRNA for KH domains through a C-terminal coiled coil, depending on its conformation. This explains why the eviction of mRNAs from KH domains required multivalent ANKS3 interactions (**Fig 7**). Specifically, we found that the SAM:SAM interaction between ANKS3 and Bicc1 significantly decreased the access of RNA, suggesting that SAM anchoring stabilizes the inhibitory conformation of the ANKS3 Cter. By contrast, cooperation of the Nter domain of ANKS3 with ANKS6 destabilized it by clamping down the coiled coil in an alternative conformation away from the RNA-binding sites. Steric hindrance by the ANKS3 Cter and the effect of ANKS6 as an agonist of allosteric activation by the ANKS3 Nter provide sophisticated leverage to license the access of Bicc1 to mRNAs. The advantage of such a system is that tissue-specific changes in the stoichiometry among these interacting factors will allow to modulate the threshold of Bicc1 binding to specific target mRNAs, and without affecting Bicc1 RNP dynamics in cells or tissues where ANKS3 is absent. To our knowledge, similar licensing of mRNA binding to a protein by mutually antagonistic structured domains in a multivalent network has not been described.

Gain-of-function studies in HEK293T cells confirmed that competition by ANKS3 also diminishes Bicc1 binding to specific target mRNAs in vivo, including its own mRNA and $Dand5$ transcripts. Conversely, $Anks3$ RNAi in IMCD3 cells increased the association of endogenous Bicc1 with its own mRNA and $Adcy6$ transcripts. Interestingly, this increased binding was accompanied by a 7.4-fold increase in the $Bicc1$ mRNA expression without a corresponding increase at the protein level. This uncoupling between mRNA and protein levels suggests the existence of posttranscriptional mechanisms limiting the accumulation of Bicc1. In $Drosophila$ egg chambers, Bicc1 binding to the 5′ UTR of its own mRNA leads to autoinhibition without destabilizing the mRNA [32]. Analogous Bicc1 autoinhibition has not been reported in vertebrates where this 5′ UTR is not conserved. However, one possible scenario is

that binding of Bicc1 to other regions stabilizes its mRNA in complexes that are not translated, at least in *Anks3* depleted cells. In particular, the 3′ UTR deserves further consideration since ANKS3 diminished only the expression of endogenous *Bicc1* mRNA, but not of HA-Bicc1 where this long sequence (3 kb) was deleted. However, the *bicc1* 3′ UTR in *Drosophila* mediates robust mRNA silencing even independently of autoinhibition [32]. Therefore, and since we found that ANKS3 did not alter Bicc1 expression at the protein level either in cultured cells or in vivo, we here focused instead on a new role of ANKS3 and its inhibition by ANKS6 in regulating the binding of Bicc1 to target RNAs.

In our loss-of-function study in IMCD3 cells, we observed that, despite its enhanced binding to Bicc1, the *Adcy6* mRNA expression level remained unchanged upon ANKS3 depletion. Likewise, in kidneys, Bicc1 represses *Adcy6* expression independently of deadenylation and decay [9]. Previously, we have shown that Bicc1 cooperates with the CCR4-NOT complex to promote the decay of *Dand5* mRNA that is induced by flow-stimulated cilia on the left side of the node [5]. However, in cultured cells lacking cilia, and in the absence of flow stimulation, Bicc1 clearly binds these target mRNAs without reducing their stability. How the decay of Bicc1-associated mRNAs is differentially regulated by their context and whether it influences the exchange of client RNAs or their competition for Bicc1 KH domains warrants further studies.

The novel inhibition of ANKS3 by ANKS6 agrees with known functions of ANKS6 during left-right development and in renal tubule homeostasis [33]. ANKS6 deficiency in *streaker* mice causes heterotaxia marked by right isomerism of the lung and sometimes of the heart atria [18], as expected if Nodal signaling is impaired by ectopic accumulation of Dand5. Moreover, a chemically induced point mutation in the SAM domain of ANKS6 that impairs its recruitment to Bicc1 in cystic kidneys of ANKS6$^{I747N/I747N}$ mice has been shown to diminish the accumulation of polycystin-2 [13], a phenotype that is reminiscent of *Bicc1* loss-of-function [34]. The amount of ANKS6 that is recruited to Bicc1-ANKS3 complexes in kidneys is limited by the kinase NEK8, which phosphorylates ANKS6 to thereby promote its retention in primary cilia by Inversin [14,19]. Therefore, future studies should investigate whether the availability of ANKS6 to release Bicc1 from ANKS3 inhibition is regulated by flow stimulation of primary cilia and by its effects on ANKS6-associated factors such as Inversin and NEK8 [17].

## Materials and methods

### Ethics statement

All mouse experiments were performed under the institutional license A2016-01-6 in accordance with guidelines of the RIKEN Center for Biosystems Dynamics Research. Mice were maintained in the animal facility of the RIKEN Center for Biosystems Dynamics Research. Euthanasia was performed by cervical dislocation.

### Antibodies

Monoclonal rabbit anti-HA (Sigma H6908), monoclonal mouse anti-FLAG M2 (Sigma F3165), polyclonal rabbit anti-ANKS6 (Sigma HPA008355), and monoclonal mouse anti-γTubulin (Sigma T6557) antibodies were used for western blot analyses. Immunoprecipitation and detection of endogenous Bicc1 in IMCD3 cells were performed using custom-made affinity-purified polyclonal rabbit anti-Bicc1 antibody [10]. Polyclonal rabbit antibodies that were unsuccessfully tested to detect endogenous ANKS3 were from Proteintech (24058-1-AP), Sigma-Aldrich (HPA041409), Aviva (ARP52561_P050), Novus Biologicals (NB100-61625), and Bethyl laboratories (A301-388A).

## Plasmids

The plasmids pGEX-1λT::Bicc1-KH [9], pCMV-SPORT6::HA-Bicc1 [27], pcDNA6::V5-ANKS6 [19], pCMV6-Entry::ANKS3-Flag and pCMV6-Entry::ANKS3ΔCter-Flag [17], and pEFBOS::*Dand5-3′ UTR* [5] have been described previously. To obtain the plasmid pCMV6-Entry::ANKS3ΔNter-Flag, a PCR fragment lacking the sequence encoding amino acids 2–220 has been cloned between the KpnI and HindIII sites. To obtain the plasmid pCMV6-Entry::ANKS3ΔNter-Flag, a PCR fragment lacking the amino acids 2–220 encoding sequence has been cloned between the KpnI and HindIII sites. To obtain the plasmid pCMV6-Entry::ANKS3ΔSAM-Flag, an oligonucleotide linker has been inserted between the HindIII and BssHII sites to remove the amino acids 368–529 encoding sequence. PCR fragments of individual KH domains and full-length Bicc1 WT or mutD have been amplified from appropriate pCMV-SPORT6 constructs [10] and cloned between BamHI and XhoI sites of pGEX-1λT and pGEX-6p1, respectively. Plasmid pGBKT::ANKS3 and fusions of human BICC1 FL (full-length) cDNA, KH repeat, IVS, or SAM domain with the activation domain of GAL4 in pACT2 have been described [17]. PCR amplicons of the KH$_1$, KH$_2$, and KHL (which encompasses KHL1, KH3, and KHL1 domains) were digested with BglII and XhoI and inserted between BamHI and XhoI sites of pACT2.

## Cell lines and cell culture

IMCD3 (CRL-2123) and HEK293T (CRL-11268) cell lines were purchased from ATCC and cultured in DMEM (Sigma) supplemented with 10% FBS (Sigma), 1% GlutaMAX (Thermo Fisher Scientific), and 1% gentamicin (Thermo Fisher Scientific). To deplete ANKS3, IMCD3-sh::ANKS3 cells [29] were seeded at a density of $2 \times 10^6$ per 10 cm dish and treated with 0.25 μg/mL doxycycline. After 2 days, cells were passaged into two 15 cm dishes at a density of $5 \times 10^6$ cells per plate and doxycycline was added in fresh complete medium every 48 h for 5 days.

## Mouse strains

NDE-Hsp-dsVenus-*Dand5* 3′ UTR transgenic mice were described previously [5]. *Anks3* mutant mice were generated by CRISPR/Cas9 editing in C57BL/6J zygotes and genotyped as indicated in S1 Fig.

## Whole mount in situ hybridization and imaging of dsVenus fluorescence

WISH was performed according to standard procedures using digoxigenin-labeled riboprobes specific for Nodal mRNA [35]. The fluorescence of dsVenus at the node was imaged using an FV1000 confocal microscope (Olympus) equipped with 60× lens (UPlanSApo 60×/1.35) and quantified as described previously [5]. Briefly, in 3D images averaged from a single z-stack, ROIs were set using the threshold function in ImageJ individually on both the L and R sides. Nonspecific background signal was measured at the center of the node. The L/R ratio of dsVenus fluorescence was calculated using the equation [(Average intensity in L)—(Background in Center)] / [(Average intensity in R)—(Background in Center)]. To account for differences in the laser power and High Voltage values, the sum of the total intensity (Left + Right) was calculated using the correction coefficient recommended for the microscope.

## Immunofluorescence analysis of mouse embryos

Dissected embryos were fixed with 4% paraformaldehyde, dehydrated with methanol, and permeabilized with phosphate-buffered saline (PBS) containing 0.1% Triton X-100, followed by

overnight incubation at 4˚C with anti-Bicc1 (1:100 dilution, rabbit polyclonal, Sigma) and anti-γ-tubulin antibodies (1:100 dilution, mouse monoclonal, Sigma). Unbound antibodies were removed by washing the embryos in PBS containing 0.1% Triton X-100, followed by incubation with Alexa Fluor-conjugated secondary antibodies (Invitrogen). The node of stained embryos was excised, placed on a slide glass with silicone rubber spacers, covered with a cover glass, and imaged with an Olympus FV3000 confocal microscope. To localize Bicc1 in node cells, cryosections were prepared from immunostained node samples and imaged by super-resolution AiryScan mode using an LSM880 confocal microscope (Zeiss). For signal quantification, individual cells were identified using the Cellpose algorithm [36]. The intensity of the Bicc1 signal in arbitrary units (AUs) was measured for each cell using the Image J software. Based on extra node cells showing no Bicc1 expression, a threshold of 50 AU was used to determine the Bicc1-positive cells.

### Analysis of public single-cell RNA sequencing data

Single-cell RNA-seq data from mesendoderm cells of mouse embryos between 7.5 and 7.75 days post coitum marked by fluorescent Foxa2-Venus fusion protein expression were downloaded from Gene Expression Omnibus (GSE162534). Data preprocessing including filtering, clustering, annotation after batch correction and counts per million (CPM) normalization and log + 1 scaling were described by Scheibner and colleagues [24]. To identify node cells marked by *Foxj1* and *Bicc1* transcripts and to generate UMAP and violin plots of *Dand5*, *Anks3*, and *Anks6* mRNA expression, we interrogated this batch-corrected and CPM normalized annotated public dataset using non-batch corrected values as raw data and Jupyter Notebooks that were made publicly available by the authors at https://github.com/theislab/gastrulation_ analysis (running under Python 3.10.4). Whenever a gene was represented by at least 1 UMI, it was considered to be expressed in that cell.

### In vitro transcription

The cDNA templates were provided with the SP6 RNA polymerase promoter and with the sequence tag TGTCTGGGCAACAGGCTCAGG at their 5′ and 3′ ends, respectively, using Overlap Extension PCR primers and Phire Green Hot Start II PCR Master Mix (Thermo Fisher). After agarose gel electrophoresis, PCR amplicons were purified on NucleoSpin Gel and PCR Clean-up (Macherey-Nagel). Template DNAs of interest (500 ng each) were transcribed in vitro during 2 h at 37˚C using SP6 RNA polymerase kit (Roche), followed by a 1-h treatment with DNAse I (Roche) prior to purification of the newly synthesize RNA on Quick Spin Columns (Roche) and quantification by $OD_{600}$ measurement.

### Purification of GST fusion proteins

Fusions of GST with Bicc1 fragments in plasmid pGEX-1λT or with full-length Bicc1 in plasmid pGEX-6p1 were expressed in *E. coli* BL21 (Novagen) as described [9] and purified using glutathione-Sepharose 4B (GE Healthcare). Purification was carried out in a buffer consisting of 50 mM Tris-HCl (pH 8), 200 mM NaCl, and 1 mM dithiothreitol (DTT).

### RNA co-immunoprecipitation assay in IMCD3 and HEK293T cells

HEK293T cells were transfected with *Dand5*-3′ UTR, HA-Bicc1 and ANKS3-Flag plasmids (2 µg of each per dish), and 8 µg v5-ANKS6 using Jet-PEI (Polyplus). To increase the dosage of ANKS3 ΔCter-Flag 4 or 8 µg was transfected per dish where indicated. HEK293T cells from two 10 cm dishes, or IMCD3-sh::ANKS3 cells from two 15 cm dishes per condition, were

washed with ice-cold PBS, extracted with 20 mM Tris-HCl (pH 7.4), 2.5 mM $MgCl_2$, 100 mM NaCl, 5% glycerol, 1 mM dithiothreitol (DTT), 0.05% Nonidet P-40 (NP-40), RNasin (Promega), phosphatase inhibitors (Sigma), and protease inhibitors (Roche), by passing them 8 times through a syringe needle (no. 30). Extracts were centrifuged twice at $10,000 \times g$ for 5 min at 4°C and 5% of each set aside as controls for the "input." For immunoprecipitation, 20 μL of Protein G-sepharose beads (GE Healthcare) coated with 12 μL rabbit anti-Bicc1, or 20 μL of mouse anti-HA beads (Sigma) and preabsorbed with RNAse-free BSA (800 μg/mL) were incubated with the remainder of each extract for 2 h at 4°C on a rotating wheel, then rinsed 4 times 10 min in wash buffer (20 mM Tris-HCl (pH 7.4), 2 mM $MgCl_2$, 200 mM NaCl, 1 mM DTT, and 0.1% NP-40). While 10% of the beads were analyzed by immunoblotting as described above, the remainder and half of each input sample were subjected to phenol-chloroform extraction. After ethanol precipitation, isolated RNA was treated with RQ1 DNase (Promega) and converted to cDNA by PrimeScript Reverse Transcriptase Kit (Takara). The resulting cDNA was subjected to PCR or qPCR analysis using Phire Green Hot Start II PCR Master Mix (Promega) or GoTaq qPCR Master Mix (Promega), respectively, using the primers For-GCTGAGCATCCTAGAGGAATGC and Rev-TAAACCCATGACTGGGGGACCAT GTCTAG for the 3′ UTR of *Dand5* mRNA, or For-ACAGAGCCTCGCCTTTGCC and Rev-CTCCATGCCCAGGAAGGAAGG for *β-actin* mRNA. The amount of co-immunoprecipitated mRNA as a percentage of the input was calculated with the formula: $100 \times 2[(Ct(Input) - \log2(100/2.5) - Ct(IP)]$, where the value 2.5 represents 2.5% of the original cytoplasmic extract, and where Ct is the cycle threshold for qPCRs on input or immunoprecipitate (IP) samples. Fold enrichment was calculated relative to cells transfected with the corresponding empty vector for HA-Bicc1 and normalized relative to the amount of HA-Bicc1 bait in the IP fraction.

## Reconstitution of multiprotein complexes and RNP by GST pull-down

To reconstitute RNPs, a fluorescent DNA probe (5′-CTGAGCCTGTTGCCCAGAC-3′) carrying a 5′-Dynomics 681 dye (Microsynth AG) was pre-annealed to the complementary 3′-tag of in vitro-transcribed Dand5-3′ $UTR_{66-110}$ or Dand5-3′ $UTR_{226-270}$ RNAs by denaturation for 3 min at 98°C and renaturation for 10 min at RT. Subsequently, the labeled RNA (100 pmol) was incubated with glutathione beads coated with GST-Bicc1 fusion protein (approximately 5 to 10 pmol) during 1 h at 4°C on wheel. To assess binding of preassembled RNPs or of GST-Bicc1 fusion proteins alone to ANKS proteins, HEK293T cells cultured in 10 cm dishes were transfected with 2 μg of ANKS3-Flag and 8 μg of v5-ANKS6 and extracted as described above. Cleared extracts corresponding to one third of a 10 cm dish per binding assay were incubated for 2 h at 4°C with glutathione-Sepharose 4B beads coated with GST alone (control), or with GST-Bicc1 fusions or RNPs. Proteins that were bound to the beads were washed and analyzed by western blotting as described above. Retention of the RNA was monitored using an Odyssey CLx Infrared Imaging System (LI-COR Biosciences) by imaging the annealed fluorescent probe directly in the gel before immunoblotting of associated proteins. Binding of the GST fusions was validated by Coomassie staining of eluted proteins. GST alone was included in all experiments as a specificity control, but after prolonged migration that was required to resolve the protein of interest at the top of the gel could not be retained due to its small size.

## Yeast two-hybrid assay

Binding of ANKS3 to various domains of human BICC1 was assessed in reciprocal yeast two-hybrid assays as described [37] by fusing each as a bait to the DNA-binding domain of the GAL4 transcription factor (GAL4-BD) in plasmid pGBKT7 (Clontech) and as a prey to the

activation domain of the GAL4 transcription factor (GAL4-AD) in plasmid pACT2 (Clontech). To monitor the induction of a HIS3 reporter gene by complexes of bait and prey fusion proteins, appropriate pairs of pACT2 (LEU2) and pGBKT7 (TRP1) plasmids were transformed into haploid *S. cerevisiae* CG1945 cells (mat a; ura3-52, his3-200, ade2-101, lys2-801, trp1-901, leu2-3, 112, gal4-542, gal80-538, cyhr2, LYS2::GAL1UAS-GAL1TATA-HIS3, URA3::GAL417-mers(x3)-CYC1TATALacZ) and strain Y187 (mat α; gal4, gal80, ade2-101, his3-200, leu2-3,112, lys2-801, trp1-901, ura3-52, URA3::Gal1UAS GAL1TATA-LacZ). Diploid progeny from crossings on YPD medium and selected on Leu⁻, Trp⁻ medium were plated on Leu⁻, Trp⁻, His⁻ medium for 3 days at 30˚C to select the cells where reconstituted GAL4 AD/BD complexes induced the HIS3 reporter gene. Where indicated, 3-Amino-1, 2, 4-triazol (3-AT) was added as a competitive inhibitor of histidine synthesis to evaluate the strength of the interactions.

## Protein structure modeling

Complex structure predictions were conducted with AlphaFold2_advanced python notebook. The presented structures are the predictions ranked with the highest model confidence. The BICC1:ANKS3 complex was modeled with the full-length sequence of human ANKS3 and human BICC1. Due to memory limitation, the full-length ANKS3:ANKS6:BICC1 complex cannot be predicted with AlphaFold2. In order to avoid this issue, some segments of the structure with a per-residue confidence score (pLDDT) lower than 50 are removed: Residue 471–774 are removed in ANKS6. Residue 424–701 and residue 939 to N-term are removed in BICC1. These segments are selected because a pLDDT lower than 50 indicates low model confidence and also indicates these segments are likely to be disordered.

## Quantification and statistical analysis

All western blot signals were quantified using the Odyssey CLx scanner software ensuring that all bands were below saturation (original blots are available in the S1 Raw Images file). Signal normalization methods are described below. Statistics were calculated with the Microsoft Excel software. Error bars represent standard deviations (SD). Student's *t* test was used to calculate *p* values, with $p \leq 0.05$, $p \leq 0.01$, or $p \leq 0.001$ represented by 1, 2, or 3 asterisks, respectively. Statistical details of individual experiments can be found in the figure legends and in the S1 Raw Values file.

For the pull-down experiments, the amounts of wild-type or mutant ANKS3-Flag that associated with GST-KH or GST-Bicc1 were expressed as percentages normalized to their amounts in the corresponding crude cell extracts (input) and relative to the control condition indicated in each experiment. The amount of V5-ANKS6 associated with GST-KH was calculated following the same method. Since the retention of ANKS6 in the complexes is mediated by ANKS3, these values were further normalized relative to the amount of ANKS3-Flag (except in Fig 2G where the fold change is only normalized to input, because the control contains no ANKS3-Flag). Finally, the retention of synthetic RNA was measured by detecting the 5'-Dynomics 681 fluorescent dye using an Odyssey CLx scanner. The amount of RNA associated with GST-Bicc1 was expressed as a percentage of fluorescent signal retained on beads after normalization to the input fraction and relative to the condition with the Dand5-3' UTR RNA fragment 66–110 (**Fig 4A**) or to the condition with GST-Bicc1 WT (**Fig 4C**).

In the *Anks3* loss-of-function experiment in IMCD3-sh::ANKS3 cells, the amount of Bicc1-bound mRNA was calculated with the formula $2^{-Ct(IP)}$ (**Fig 5C**). To integrate the fluctuations in the amounts of Bicc1 immunoprecipitated between samples, the resulting values were normalized to the intensity of the Bicc1 protein signal measured by western blot in

immunoprecipitates (IP). Finally, to visualize the effect of the doxycycline-induced *Anks3* depletion, results were expressed relative to the condition without doxycycline.

For *Anks3* gain-of-function experiments in HEK293T cells, the percentage of bound/total mRNA was calculated with the formula $100 \times 2^{-Ct(IP)}/2^{-Ct(Input)}$. To take into account the fluctuations in the efficiency of Bicc1 immunoprecipitation between samples, the resulting raw values were normalized to the intensity of the Bicc1 protein signal measured by western blot in the IP fractions. The fold enrichment was calculated relative to the background level in cells transfected with the corresponding empty vector for HA-Bicc1 (**S4A Fig**). Finally, to visualize the effects of ANKS proteins, the values were normalized relative to the control condition with only HA-Bicc1 (**Figs 6A** and **S5B**). The values for the levels of immunoprecipitated ANKS3 represent the amounts of ANKS3-Flag relative to HA-Bicc1 in the IP fraction. To calculate the fold change, the values were normalized relative to the control condition with WT ANKS3-Flag alone (**Fig 6B**).

## Supporting information

**S1 Fig. Generation of *Anks3* mutant mice by CRISPR/Cas9 editing, and imaging of Bicc1 and of dsVenus-*Dand5* 3′ UTR reporter transgene expression at the node. (A)** Alternative splicing of mouse *Anks3* annotated in Ensembl [38] and coding regions of domains used to raise the commercial antibodies indicated. **(B)** Anks3 mutant mice were generated using 2 gRNAs (green bars) to delete exons 10 and 11. Positions of PCR primers (blue arrows) used for genotyping are indicated. **(C)** Fluorescence of dsVenus at the node of *Anks3*$^{\Delta/\Delta}$ embryos and control litter mates harboring the NDE-Hsp-dsVenus-*Dand5* 3′ UTR transgene (TG+) at 3- to 5-somite stages. Pictures correspond to individual values presented in the graphs of Fig 1C prior to correction for variation of laser power and voltage settings. **(D)** Immunofluorescence staining of Bicc1 (green) in node cells of *Anks3*$^{+/+}$ and *Anks3*$^{\Delta/\Delta}$ embryo at E8.0. Quantification of Bicc1 immunofluorescence intensities is shown in the graph on the right as the means + SD from 112 and 161 cells from 4 *Anks3*$^{+/+}$ and 5 *Anks3*$^{\Delta/\Delta}$ animals, respectively. ns: nonsignificant (Student's *t* test). **(E)** UMAP plots showing the expression of the node cell markers *Foxj1* and *Bicc1* ($n = 1,363$ cells), corresponding to the axial mesendoderm population in a public scRNA-seq dataset [24]. Cell containing at least 1 unique molecular identifier corresponding to Bicc1 were considered to be Bicc1+. *Dand5*, *Anks3*, and *Anks6* mRNA expression is also observed in the axial mesendoderm population (shown in UMAPs by the square box and represented by violin plots). Underlying data can be found in the S1 Raw Values file. (TIF)

**S2 Fig. Structural modeling of Bicc1, ANKS3, and ANKS6. (A)** Model structures of the indicated proteins released by AlphaFold. Structured domains are annotated. Serine/glycine (SG)-rich linker regions, including the intervening sequence (IVS) of Bicc1 are predicted to be intrinsically disordered. **(B)** Cartoon depicting the multivalent interactions in the Bicc1-ANKS3-ANKS6 protein network. Connecting straight lines indicate validated protein–protein interactions. The one-way sign indicates that the EH surface of the ANKS6 does not bind the SAM domains of either Bicc1 or ANKS3. (TIF)

**S3 Fig. Expression of *Bicc1* and of specific target mRNAs in IMCD3-sh::Anks3 cells. (A)** RT-PCR detection of endogenous mRNAs in the input fraction of untreated IMCD3-sh::ANKS3 cells used for co-immunoprecipitation in Fig 5. cDNA from mouse mammary gland was used as a positive control for the expression of *Dand5* based on a search for *DAND5*-expressing cell lines and tissues in the Human Protein Atlas database (https://www.

proteinatlas.org/ENSG00000179284-DAND5/tissue). **(B)** Expression levels of *Bicc1* and *Adcy6* mRNAs relative to *β-actin* measured by RT-qPCR analysis in IMCD3 before and after doxycycline-induced *Anks3* depletion. The values are expressed as the fold change normalized to the untreated condition. These data are related to the co-immunoprecipitation results shown in Fig 5. Data are means + SD from 4 independent experiments. ns: nonsignificant, $^*p < 0.05$, $^{**}p < 0.01$, $^{***}p < 0.001$ (Student's *t* test). Underlying data can be found in the S1 Raw Images and S1 Raw Values files.
(TIF)

**S4 Fig. ANKS3 and ANKS6 modulate the binding of target mRNAs to HA-Bicc1 in HEK293T cells. (A)** Additional analysis of the RT-qPCR data shown in Fig 6B. Ratios of co-immunoprecipitated mRNAs over the input, normalized to the amount of HA-Bicc1 in the IP fraction and then expressed relative to the corresponding control condition without HA-Bicc1. *β-actin* mRNA served as a negative control to visualize the level of unspecific RNA binding by Bicc1. **(B)** Expression level of the dsVenus-*Dand5* 3′ UTR and *HA-Bicc1* mRNAs relative to *β-actin* measured by RT-qPCR analysis in transfected HEK293T cells. These data are related to the co-immunoprecipitation results shown in Fig 6B. Data are means + SD from between 3 to 10 independent experiments. ns: nonsignificant, $^*p < 0.05$, $^{**}p < 0.01$, $^{***}p < 0.001$ (Student's *t* test). Underlying data can be found in the S1 Raw Values file.
(TIF)

**S1 Table. Phenotypes in *Anks3*$^{\Delta/\Delta}$ mutant mice.**
(XLSX)

**S2 Table. Oligonucleotide primer sequences used in this study.**
(PDF)

**S1 Raw Images. Original blot images contained in main and supplementary figures.**
(PDF)

**S1 Raw Values. Reporting of individual data points and statistical analysis shown in main and supplementary figures.**
(XLSX)

## Acknowledgments

The authors would like to thank Dr. Gerd Walz for kindly providing the IMCD3-sh::ANKS3 cell line and Dr. Soeren Lienkamp for v5-ANKS6 plasmid. We are grateful to Dr. Maren Büttner, Institute of Computational Biology, Helmholtz Zentrum München, Munich, Germany, for sharing filtered scRNA-seq datasets and python notebooks to help with our re-analysis. We also are indebted to Dr. Cathrin Brisken (EPFL) for providing cDNA from adult mouse mammary glands. This work was supported by resources and services of the Bioimaging Research Core Facility and the Gene Expression Core Facility at the School of Life Sciences of EPFL.

## Author Contributions

**Conceptualization:** Benjamin Rothé, Yayoi Ikawa, Zhidian Zhang, Takanobu A. Katoh, Eriko Kajikawa, Katsura Minegishi, Sai Xiaorei, Simon Fortier, Hiroshi Hamada, Daniel B. Constam.

**Data curation:** Benjamin Rothé.

**Formal analysis:** Benjamin Rothé, Yayoi Ikawa, Zhidian Zhang, Takanobu A. Katoh, Eriko Kajikawa, Katsura Minegishi, Sai Xiaorei, Simon Fortier.

**Funding acquisition:** Simon Fortier, Daniel B. Constam.

**Investigation:** Benjamin Rothé, Yayoi Ikawa, Zhidian Zhang, Takanobu A. Katoh, Eriko Kajikawa, Katsura Minegishi, Sai Xiaorei, Simon Fortier.

**Methodology:** Benjamin Rothé, Yayoi Ikawa, Zhidian Zhang, Takanobu A. Katoh, Eriko Kajikawa, Katsura Minegishi, Sai Xiaorei, Simon Fortier, Hiroshi Hamada, Daniel B. Constam.

**Project administration:** Daniel B. Constam.

**Supervision:** Daniel B. Constam.

**Validation:** Benjamin Rothé, Daniel B. Constam.

**Visualization:** Benjamin Rothé.

**Writing – original draft:** Benjamin Rothé, Daniel B. Constam.

**Writing – review & editing:** Benjamin Rothé, Zhidian Zhang, Simon Fortier, Matteo Dal Peraro, Hiroshi Hamada, Daniel B. Constam.

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
