## [Editor Report · Decision Letter 0]

1 Feb 2023

Dear Dr Constam, 

Thank you for submitting your manuscript entitled "Mutually antagonistic interactions among structured domains in the multivalent Bicc1-ANKS3-ANKS6 protein network control binding of target mRNAs." for consideration as a Research Article by PLOS Biology. Please accept my apologies for the delay in getting back to you as we consulted with an academic editor about your submission. 

Your manuscript has now been evaluated by the PLOS Biology editorial staff, as well as by an academic editor with relevant expertise, and I am writing to let you know that we would like to send your submission out for external peer review.

Once your full submission is complete, your paper will undergo a series of checks in preparation for peer review. After your manuscript has passed the checks it will be sent out for review. To provide the metadata for your submission, please Login to Editorial Manager (https://www.editorialmanager.com/pbiology) within two working days, i.e. by Feb 03 2023 11:59PM.

Kind regards,

Richard

Richard Hodge, PhD

Associate Editor, PLOS Biology

rhodge@plos.org

PLOS

---

## [Decision Letter · Decision Letter 1]

23 Mar 2023

Dear Dr Constam,

Thank you for your patience while your manuscript "Mutually antagonistic interactions among structured domains in the multivalent Bicc1-ANKS3-ANKS6 protein network control binding of target mRNAs" was peer-reviewed at PLOS Biology. Please accept my sincere apologies for the delays that you have experienced during the peer review process. It has now been evaluated by the PLOS Biology editors, an Academic Editor with relevant expertise, and by three independent reviewers. 

In light of the reviews, which you will find at the end of this email, we would like to invite you to revise the work to thoroughly address the reviewers' reports.

As you will see, the reviewers think that your study is interesting and well done, but they raise overlapping concerns with the strength of the data demonstrating the physiological significance of the Bicc1-ANKS3-ANKS6 crosstalk. The reviewers provide several experimental suggestions that we agree should be included to further demonstrate the in vivo relevance of the proposed mechanism. In addition, they raise concerns with the reporting and analysis of the single cell and RIP loss-of-function studies. 

Given the extent of revision needed, we cannot make a decision about publication until we have seen the revised manuscript and your response to the reviewers' comments. Your revised manuscript is likely to be sent for further evaluation by all or a subset of the reviewers.

**IMPORTANT - SUBMITTING YOUR REVISION**

*Re-submission Checklist*

*Published Peer Review*

*PLOS Data Policy*

*Blot and Gel Data Policy*

Sincerely,

Richard

Richard Hodge, PhD

Associate Editor, PLOS Biology

rhodge@plos.org

REVIEWS:

Reviewer #1: This manuscript by Rothe, et al provides novel insights into the regulation of the RNA binding protein Bicc1, which has been implicated in mediating ciliopathy-associated phenotypes in animal models. The study focuses on the previously identified Bicc1 interacting proteins Anks3 and Anks6. Anks3 knockout mice show reduced expression of the Bicc1 target Dand5, and subsequently develop laterality defects. These results suggest Anks3 antagonizes Bicc1 activity. The authors then use modeling and multiple biochemical approaches to interrogate the interactions between Bicc1, Anks3, Anks6, and mRNA. Convincing results from these experiments support the authors' model that Anks3 inhibits Bicc1 from binding target mRNAs and that recruitment of Anks6 alters Anks3 conformation to promote Bicc1 interaction with mRNA. The authors propose an intriguing testable model in which availability of Anks6 that modulates Bicc1 is regulated via localization/sequestration in primary cilia. Overall, the manuscript is well written and the data are of high quality and clearly presented. The use of statistical analysis is appropriate. The findings are significant in that they shed light on molecular mechanisms of mRNA regulation in contexts that are relevant to human diseases. I have only minor comments for the authors to consider:

1) In the first paragraph of the Introduction section, I suggest citing recent work from the Hamada lab (Katoh et al, Immotile cilia mechanically sense the direction of fluid flow for left-right determination. Science. 2023 Jan 6;379(6627):66-71) to provide additional support for the statement "…stimulation of primary cilia by a leftward fluid flow accelerates the decay of Dand5 mRNA..."

2) In the second paragraph of the Introduction section, I suggest spelling out sterile α motif (SAM), which currently appears in the third paragraph. 

3) From the analysis of laterality defects in Anks3 knockout mice, it is implied that only homozygous mutants present laterality defects since heterozygous animals were included in 'controls.' If indeed laterality defects were not observed in heterozygous mice, this should be clearly stated. On the other hand, a low incidence of laterality defects would also be interesting. 

4) In the Results section titled 'Endogenous ANKS3 buffers the binding of Bicc1 to specific target mRNAs in IMCD3 cells,' I suggest changing 'in vivo' to 'in cells' in the sentence "To test whether ANKS3 and RNA compete for Bicc1 binding in vivo, we analyzed endogenous Bicc1 RNPs…"

Reviewer #2: The manuscript by Rothé et al., presents embryological and biochemical data on the regulation of the posttranscriptional properties of Bicc1 by Anks3 and Anks6 with respect to Dand5 mRNA. Like in other vertebrate species, Anks3 loss-of-function in mice resulted into impaired organ laterality, which was caused by failed LR patterning. In Anks3 mutants, Nodal expression in the left lateral plate mesoderm was either lost or ectopically induced on the right. Anks3 function in laterality determination was further pinpointed to the process of symmetry breakage at the node. Judged by a transgene, Bicc1 dependent Dand5 asymmetry, which is the consequence of cilia-driven leftward flow, was no longer observed. The authors interpreted this result as precocious symmetric decay of Dand5 mRNA which was triggered by Bicc1. Therefore, Anks3 could serve as a Bicc1 regulator during symmetry breakage. Using structure prediction by Alphafold and numerous complex biochemical experiments, the authors dissect the molecular basis of Bicc1 interactions with Anks3 and Dand5 3'UTR. Surprisingly, the RNA binding domain of Bicc1 specifically interacts with the C terminus of Anks3 and thereby interfered with Dand5 mRNA binding. Further, the authors show that inhibition of RNA binding to Bicc1 by Anks3 was prevented by the presence of Anks6. A Bicc1 / Anks3 / Anks6 interplay had been established by the Constam group in the context of Bicc1 polymerization into granules. The new data now shows the effect of Bicc1 with respect to one of its target mRNAs. As such the manuscript is highly relevant and would fit to the scope of PLOS Biology. However, I do have some major and minor issues, which need to be addressed. 

Major issues:

- The fundamental problem of the manuscript lies in the disconnect of embryological and biochemical data, which impede a coherent story. The analysis of the Anks3 mouse mutant provides no experimental evidence which underscores or matches with the demonstrated mechanism of Bicc1 / Ansk3 / Anks6 interaction. Moreover, the presented mouse data can be considered as preliminary and some of the conclusions are at least questionable.

1. As the authors noted, it is not clear if the CRISPR introduced deletion results into a null allele. Because the retained N-terminal sequence of Anks3 contains the ANK domain, a truncated protein could be expressed and functional the phenotype of a "true" knockout could differ. The lack of a kidney phenotype is an indication of retained Anks3 functions in the CRISPR mutant. However, clarifying this issue would be beyond the scope of the manuscript. The authors could test the corresponding N-terminal Anks3 sequence in their experimental setups and thereby provide some evidence on the potential activity of the mutant protein. In any case, the authors should take this circumstance into account and should not label the homozygous mutants as knockouts.

2. The authors suggest that the Anks3 mutants somewhat phenocopies Dand5 knockout embryos. However, Dand5 knockout embryos show higher frequency of bilateral Nodal expression and prolonged Nodal expression was not reported. Later observation in Anks3 mutants may hint to reduced Lefty expression and/or disrupted midline. In consequence, Anks3 likely has more functions during LR development.

3. In Anks3 mutants, precocious symmetric decay of Dand5 mRNA was suggested to be the cause of LR defects. This reasoning was deduced from a transgene, which drives expression of a destabilized Venus protein in the crown cells of the node. The corresponding transcript of the transgene is under the control of the Dand5 3'UTR. Besides representing only an indirect evidence, the timepoint of the analysis was too late to draw that specific conclusion. At 3-5 somite stage embryos, leftward flow was already sensed, which is demonstrated by Dand5 asymmetry. Hence, bilateral reduction of Dand5 could be the cause of a turbulent flow, which triggers Dand5 inhibition / decay in left and right crown cells (see 4.). Precocious Dand5 decay should be detectable at stages where leftward flow was not established yet or just started. Based on experiments of the Hamada group, first Dand5 asymmetries are already detectable at early headfold (EHF) stage. Dand5 asymmetries are basically penetrant as soon as somites form. In addition, blocking flow with methylcellulose at different stages, revealed that flow at EHF, which is weak and slow, is sufficient to determine laterality (Shinonara et al., 2011). Therefore, analysis of endogenous Dand5 mRNA expression prior somite stages needs to be presented to proof precocious symmetric decay. 

4. Anks3 and 6 have a record of being bona fide ciliopathy genes. Loss of Anks6 can alter cilia polarity. Planar polarity of ciliated LRO cells is a prerequisite to generate a directional flow. Impaired cilia polarity can create turbulent flow, which could lead to symmetric Dand5 downregulation. Therefore, cilia parameter like length and polarity needs to be assessed in Anks3 mutants. Flow analysis could provide data which solves this question, as well. In this context, Bicc1 could still be the critical target for Anks3, as Bicc1 loss-of-function impacts on cilia polarity, as well. 

- Posttranscriptional autoregulation by Bicc1 is an interesting feature, although a connection to the overall story is not obvious. An established Bicc1 target mRNA would be more qualified to be experimentally compared to the Dand5 3'UTR. 

If I got it correctly, HA-Bicc1 binds to its own mRNA. However, because the HA-Bicc1 expression vector does not contain the Bicc1 3'UTR, HA-Bicc1 binding must occur via the coding sequence of its mRNA. To my knowledge, Bicc1 generally acts via the 3'UTRs of target mRNAs. Thus, this observation is intriguing and I was wondering if something related has been reported before. In addition, this result raises the question to which extent such Bicc1 binding to its own coding sequence is functionally relevant and whether there is an general interference in Bicc1 gain-of-function experiments. I envision that HA-Bicc1 mRNA serves as a constant competitor to other mRNA targets i.e. Dand5 3'UTR. Does this issue has an impact on the obtained results? 

Which sequence in HA-Bicc1 mRNA gets bound by HA-Bicc1?

- Recently, the authors have shown an Bicc1 / Anks3 / Anks6 interaction in the context of Bicc1 granulaes. How does granulae formation contribute to the observations on Dand5 3'UTR binding by Bicc1? The mutant Bicc1bks mouse line does not display LR patterning defects and this mutant Bicc1 protein is unable to polymerize. Is this mutant protein behaving identical to wt Bicc1 when used in the biochemical assays? Such data set would establish a stronger link to the formation of the LR axis. 

Minor points 

- Italic letters for genes and transcripts needs to be applied throughout the text. Currently, italic style for mRNAs / genes is rather randomly used. 

- Figure 3B: The nature of asterisk is not easily recognizable. The artefact could be directly marked. 

- The different normalizations and calculations are difficult to follow. More detailed explanations in figure legends or in the method section would be surely helpful. 

- The single cell data of early mouse embryos (Figure 1D) should be viewed as circumstantial evidence and removed from the main figure. The existence of Foxj1+ / Bicc1- / Dand5 + cells is particularly odd, as the authors show uniform Bicc1 expression in the node (Figure S1B). Bicc1 violine is shifted upwards. 

- Is Anks3 expression detectable in node cells by in situ hybridization?

- Figure S3: I am not aware of a report showing Dand5 expression in mammary glands. Please add reference when I am wrong. 

Reviewer #3: The manuscript by Rothe et al. provides new mechanistic and biological insights into a crosstalk between Anks3, Anks6 and Bicc1. In particular the mouse data and the competition experiments are very compelling using overexpressed proteins are very convincing. Unfortunately, the loss-of-function studies and thus the in vivo significance of this crosstalk are not as well developed. Thus, whether finetuning of the levels of Anks3 and Anks6 is biologically critical is still lacing. In addition, there are several other aspects of the manuscript that require improvement.

Major Points:

A. Single Cell Studies: 

1. The single cell analysis in Figure 1D must include feature plots for Anks6, Anks3 and Dand5. Only by including the feature plots can the reader really judge whether they are co-expressed. In fact, it is surprising that the authors have not provided a more traditional in situ-based support for the co-expression of these 4 RNAs/proteins using e.g. RNAscope.

2. The violin plots do not support the statement of the authors that Anks6 and Dand5 are higher in the Bicc1- cells. They appear very similar. Statistics need to be performed to support such a statement.

3. The subclustering needs to be interpreted with caution. The annotation of the single cell cluster under investigation is "axial mesendoderm" and not all of the cells in this cluster are Foxj1+/Bicc1+crown cells. The Foxj1+/Bicc1- cells may be of a completely different types and thus the authors would be comparing two unrelated cell types. Thus, this comparison needs to be removed from the manuscript, or supported by a more in-depth validation by e.g. RNAscope. 

4. Finally, the methods section on the single cell data analysis needs to be expanded so that the reader can adequately judge how the authors reached their conclusions. Just referring to the Github page is insufficient. 

B. RIP Loss-of-Function Studies: 

1. In Figure 5C and D are confusing. They are the same data but normalized in different ways. This raises the question about the authors' interpretation. Providing two options, suggests that the "corrections" influence the data. Thus, the authors need to show the raw data of this experiment, i.e. the non-normalized and not corrected levels in the presence (one column) or absence of Dox (second column) for each of the three genes of interest (Bicc1, Adcyd6 and Ddx5) and the b-Actin control in the main figure. It is important that the readers can judge the unmanipulated results. Finally, while there is value in the "corrected" data, the authors need to decide, which one of the manipulations yields the most accurate representation of the data and this should be part of the main figure (in addition to the raw data). The other panel should be relegated to the supplement. Finally, it is unclear why the authors normalize to the amount of IP'ed Bicc1 protein. In the description of the previous panel, they stated that "[…] without changing Bicc1 protein levels (Fig. 5B)." Thus, Bicc1 levels should not be a factor and no correction for this is needed. 

2. In the same paragraph, the final conclusion appears contradictory in respect to Adcy6. The authors state "By contrast, the accumulation of Adcy6 mRNA was unaffected, consistent with previous observations that it also remains unchanged upon binding to Bicc1 (Piazzon et al., 2012)." Then in the next sentence the state "Taken together, these results show that ANKS3 attenuates Bicc1 binding to Adcy6 and Ddx5 transcripts […]." So, is Adcy6 increased or not.

C. Anks3 Antagonism studies.

1. In Figure 6 the authors investigate a "possible antagonism between the effects of Nter and Cter regions of ANKS3 on the RNA binding activity of Bicc1. While the different activities of the two parts of the proteins are very clear, the Anks6 dCter is expressed at lower levels than the dNter (Fig. 6B). To rule out that the effect is more due to expression levels, the authors should perform the experiments with different concentrations of the Anks6 mutation constructs. 

D. Experimental Details.

The authors need to provide more experimental detail on the amounts of constructs/proteins used. As the entire study is about stoichiometry, this information is critical for the data interpretation. The authors need to thoroughly go through the entire manuscript and amend the experimental details either in the Methods section or the figure legends. 

Minor Points:

E. The statement in the discussion "Cystic growth in kidneys is stimulated by cAMP (Wallace, 2011), which also accumulates in Bicc1 mutant kidneys, likely due to de- repression of adenylate cyclase 6 (Adcy6) mRNA translation (Piazzon et al., 2012)." is an oversimplification of PKD. While Ca2+ plays a role it, it by far not the only pathway dysregulated in ADPKD. Thus, the authors should change this in lieu of one acknowledging the complexity of signaling impacted in PKD.

F. The authors state that "Anks3-/- neonates showed no kidney cysts (unpublished observation), […]" These data should be included in the manuscript in the supplemental material, as this is important for the fact that the crosstalk with Ankst6 and Bicc1 during L/R specification and kidney disease are different.

---

## [Decision Letter · Decision Letter 2]

2 Aug 2023

Dear Dr Constam,

Thank you for your patience while we considered your revised manuscript "Mutually antagonistic interactions among structured domains in the multivalent Bicc1-ANKS3-ANKS6 protein network license mRNA binding." for publication as a Research Article at PLOS Biology. Please accept my apologies for the delays that you have experienced during this round of the peer review process. This revised version of your manuscript has been evaluated by the PLOS Biology editors, the Academic Editor and the original reviewers.

Based on the reviews, I am pleased to say that we are likely to accept this manuscript for publication, provided you satisfactorily address the remaining points raised by the reviewers. This includes discussing alternative interpretations of the findings and toning down some statements referring to the single cell data in the manuscript text. Please also make sure to address the following data and other policy-related requests that I have provided below (A-E):

(A) We would like to suggest the following modification to the title to make it more broadly accessible to general readership:

““ANKS6 modulates the conformation of ANKS3-Bicc1 complexes and licenses their recruitment to specific transcripts to regulate organ laterality”

(B) In the ethics statement in the Methods section, please include the full name of the IACUC/ethics committee that reviewed and approved the animal care and use protocol. Please also include the method of euthanasia used to sacrifice the mice.

(C) Please also ensure that each of the relevant figure legends in your manuscript include information on *WHERE THE UNDERLYING DATA CAN BE FOUND*, and ensure your supplemental data file/s has a legend.

(D) Thank you for already providing the original ans uncropped images for the western blots presented in the manuscript. However, we note that the original images for Figure S3B has not been provided in the raw image file, so we ask that this is included at this stage.

(E) Please ensure that your Data Statement in the submission system accurately describes where your data can be found and is in final format, as it will be published as written there. 

We expect to receive your revised manuscript within two weeks. 

*Published Peer Review History*

*Press*

Sincerely,

Richard

Richard Hodge, PhD

rhodge@plos.org

Reviewer remarks:

Reviewer #1: The authors have addressed all of my comments on the manuscript and have improved clarity. I find the revised manuscript appropriate for the audience of PLoS Biology. 

Reviewer #2: The authors addressed most of my questions and based on the convincing biochemistry I support the publication in PLOS Biology.

In my view, one issue remains. The sole analysis of the Dand5 3'UTR transgene in Anks3 mutants is not sufficient to conclude that the mechanistic cause of LR defects is based on precocious decay of dand5 mRNA. Loss of transgene asymmetry and reduction of signals, at embryonic stages were flow has been already sensed, could be the consequence of two possible modes of Anks3 function: 1. flow independent, precocious decay of dand5 mRNA, as the authors propose and / or 2. turbulent flow, which results into bidirectional fluid motion and thus flow sensing at the right and left side of the node. 

The reasoning for the second scenario is based on:

Turbulent flow has been reported in mouse and frog. Song et al., 2010 demonstrated that Wnt/PCP mutants have lost cilia polarity which resulted in a turbulent flow at the node and to left isomerism in mutant mice. Another example of turbulent flow was detected in Inv/Inv mutants. As stated by the authors, the development of Situs inversus in Inv/Inv mutants seems not congruent with flow pattern. However, Oki et al., 2009 demonstrated that the genetic background contributed to the phenomena. In a hybrid background, Inv/Inv mutants depict mostly bilateral activation of the nodal cascade, which matches with flow pattern. Most importantly, Bicc1 mutants also show defective cilia polarity and turbulent flow. 

Based on the published record, a hypothesis where Bicc1 interacting proteins like Anks3 or 6 are required for cilia polarity and thus for directionality of leftward flow is reasonable. 

The current analysis of the Dand5 3'UTR transgene in Anks3 mutants is not able to discriminate between both scenarios because predicted results are identical, i.e. loss of asymmetry and reduction of expression on the right. 

The strong biochemical data, provided by the authors, suggests that precocious Dand5 decay in Anks3 mutants is indeed very likely but still the authors should tune down their wording about the in vivo situation. The acknowledgement and description of an alternative scenario does not diminish the relevance of the work and just reflects the complexity of Bicc1 functions during LR development. 

Reviewer #3 (Oliver Wessely, signs review): For the revised manuscript the authors have been very responsive to the reviewers comment. As a result the manuscript has been vastly improved. I still have two comments on the single cell data, which need to be addressed.

In response to the comments on the single cell data, the authors changed the text to "We found that Anks3, Anks6 and Dand5 are indeed transcribed in the Foxj1+/Bicc1+ cells (Fig. S1E)." Yet, this wording suggests that these genes are specifically expressed in the the Bicc1-positive cells, which obviously is not the case. Thus, the authors should rephrase this sentence to something like "While Anks3, Anks6 and Dand5 are broadly expressed in the developing embryo, they are also expressed in the Foxj1+/Bicc1+ cells (Fig. S1E). They also should remove "likely" from the next sentence, because the scRNA-seq data do not provide a strong reason to perform the subsequent experiment, but instead support the author's hypothesis that this is a possible interaction. Finally, the methods section, is still very general and should be modified to actually describe how the panels, which are part of Fisgure S1E were generated.

---

## [Editor Report · Decision Letter 3]

17 Aug 2023

Dear Dr Constam,

Thank you for the submission of your revised Research Article "Bicc1 ribonucleoprotein complexes specifying organ laterality are licensed by ANKS6-induced structural remodeling of associated ANKS3" for publication in PLOS Biology. On behalf of my colleagues and the Academic Editor, Cecilia Lo, I am pleased to say that we can accept your manuscript for publication, provided you address any remaining formatting and reporting issues. These will be detailed in an email you should receive within 2-3 business days from our colleagues in the journal operations team; no action is required from you until then. Please note that we will not be able to formally accept your manuscript and schedule it for publication until you have completed any requested changes.

PRESS

Best wishes, 

Richard

Richard Hodge, PhD

rhodge@plos.org

PLOS
